# Experimental evidence of the rear capture of aerosol particles by raindrops

Pascal Lemaitre[1], Arnaud Querel[2*], Marie Monier[3,4], Thibaut Menard[5], Emmanuel Porcheron[1], Andrea I. Flossmann[3,4]

[1] Institut de Radioprotection et de Sûreté Nucléaire (IRSN), PSN-RES, SCA, LPMA, Gif-sur-Yvette, France

[2] Institut de Radioprotection et de Sûreté Nucléaire (IRSN), PRP-CRI, SESUC, BMCA, Fontenay-aux-Roses, France

[3] Clermont Université, Université Blaise Pascal, Laboratoire de Météorologie Physique, Clermont-Ferrand, France

[4] CNRS, INSU, UMR 6016, LaMP, Aubière, France

[5] CNRS UMR 6614 - CORIA Rouen, Site Universitaire du Madrillet, Saint Etienne du Rouvray, France

[*] Currently Strathom Energie, Paris, France

Correspondence to: P. Lemaitre (pascal.lemaitre@irsn.fr)

**Abstract.** This article presents new measurements of the efficiency with which aerosol particles of accumulation mode size are collected by a 1.25 mm sized raindrop. These laboratory measurements provide the link to reconcile the scavenging coefficients obtained from theoretical approaches with those from experimental studies. We provide here experimental proof of the rear capture mechanism in the flow around drops, which has a fundamental effect on sub-microscopic particles. These experiments thus confirm the efficiencies theoretically simulated by Beard (1974). Finally, we propose a semi-analytical expression to take into account this essential mechanism to calculate the collection efficiency for drops within the rain size range.

## Introduction

Aerosol particles are important components of the atmosphere. They contribute significantly to the Earth's energy budget by interacting with solar radiation, directly as well as indirectly by serving as precursors to cloud formation (Cloud Condensation Nuclei, or CCN) which also will interact with this radiation (Twomey, 1974). Furthermore, the physical properties of these particles in suspension within the atmosphere (size, concentration, affinity for water, etc.) are essential parameters for characterizing air quality. For these reasons, the scientific community has actively studied the physics of atmospheric aerosol particles.

Aerosol particles origins are several. The primary natural sources are sea spray, wind-driven dust, volcanic eruptions and human activities. The secondary sources are associated with the gas-to-particle conversion of

certain gases present in the atmosphere. The size of these particles greatly varies and ranges from one nanometre to several hundred microns. Particles of anthropogenic origin represent an increasingly large proportion of aerosol particles in the atmosphere (Charlson *et al*., 1992, Wang *et al.,* 2014). Among all anthropogenic pollutants, radioactive releases from a nuclear accident are of high risks for both humans and environment. Just like all other particles, once emitted, radioactive particles undergo physical processes that drastically change their size distribution during their transport in the atmosphere. Ultrafine particles are very sensitive to Brownian diffusion and grow by coagulation. Large particles settle on the ground due to gravity. Hence, there is a particle size range that has no efficient removal process and that has a very long atmospheric residence time. This size range is referred to as the accumulation mode (Whitby, 1973) and comprises particles with a diameter between 0.1 μm and 2 μm. These particles can remain in the upper troposphere for several months (Jaenicke, 1988) and can be transported over long distances, crossing oceans and continents (Pruppacher *et al.*, 1998).

The accumulation of particles within this size range is essentially limited by two atmospheric processes: in-cloud scavenging (rainout) and below-cloud scavenging (washout) during rainfall events. Thus, in the event of a nuclear accident with a release of radioactive aerosol particles, it is essential to correctly model both of these mechanisms in order to predict their number concentration within the troposphere as well as the ground contamination.

This study focuses on the below-cloud scavenging of aerosol particles by rain with a microphysical approach. We aim to measure in laboratory the collection efficiency of the aerosol particles constituting the accumulation mode, by drops of a size representative of rain. Recent measurements with 2 mm drops (Quérel *et al.,* 2014b) have shown that, for submicronic particles the collection efficiency increases very rapidly when the size of the particles is reduced. The Slinn (1977) model does not reproduce this increase in efficiency, leading to errors of several orders of magnitude for the collection efficiency. We impute this discrepancy to the key hypothesis of the Slinn model which assumes Stokes flow conditions around the drop. Yet, since the Reynolds number of a 2 mm drop at its terminal velocity is approximately 800, this assumption of Stokes flow is unjustified. This model nonetheless remains the most common in the literature mainly because it is easy to use.

Quérel *et al*. (2014b) showed that the Beard (1974) model was the only one to predict this increase in the collection efficiency for submicronic aerosol particles. However, direct measurements in the drop size range simulated by Beard (1974) could not be performed, the only comparison results from a linear extrapolation of theoretical computations to the measured size range. These efficiencies compared reasonably well even for aerosol particles in the submicron range. But the linear extrapolation is not completely satisfactory for an experimental validation of this model. This article provides experimental evidence of the robustness of K.V. Beard's simulation for the raindrop sizes under investigation in his paper, i.e. for diameters between 0.28 and 1.25 mm.

Our paper is divided into three sections. First, we present a theoretical description of aerosol scavenging by rain. We then present our experimental setup and the associated experimental results. Finally, we compare our measurement results with the outcomes of the models of Beard (1974) and Slinn (1977) in order to propose a semi-empirical correlation for calculating the elementary collection efficiency associated with rear capture.

40

## 1    Theoretical description of washout

At mesoscale, the scavenging of aerosol particles by rain is described by the scavenging coefficient (λ). This parameter is defined as the fraction of particles of diameter $d_{ap}$ captured by the raindrops per unit time (eq. 1). In this equation $C(d_{ap})$ is the concentration of aerosol particles of diameter $d_{ap}$ in suspension in air per unit volume.

$$\frac{dC(d_{ap})}{C(d_{ap})} = -\lambda_{rain}(d_{ap})dt \qquad (1)$$

This parameter is essential for predicting air quality (Chate, 2005) and ground contamination following a nuclear accident with release of radionuclides into the environment (Groëll *et al.,* 2014; Quérel *et al.,* 2015). There are several approaches for determining this parameter. It can either be determined theoretically by solving equation (2) (Flossmann, 1986; Mircea *et al.*, 1998; 2000) or measured in the environment by monitoring the variation of particulate concentration in the atmosphere during precipitation (Volken & Schumann, 1993; Laakso *et al.,* 2003; Chate, 2005; Depuydt, 2013 ).

$\lambda_{rain}(d_{ap}, D_{drop})$ is defined by :

$$\lambda_{rain}(d_{ap}, D_{drop}) = \int_{D_{drop}=0}^{\infty} \frac{\pi D_{drop}^2}{4} \cdot U_{\infty}(D_{drop}) E(d_{ap}, D_{drop}, RH) N(D_{drop}) dD_{drop} \,, \quad (2)$$

$D_{drop}$ is the drop diameter, $U_{\infty}(D_{drop})$ is the terminal fall velocity, $N(D_{drop})dD_{drop}$ is the number concentration of drops with a diameter between $D_{drop}$ and $D_{drop} + dD_{drop}$ during the rainfall event, $E(d_{ap}, D_{drop}, RH)$ is the collection efficiency for a given drop size, particle size ($d_{ap}$), and relative humidity (*RH*).

Unfortunately, these two approaches yield $\lambda_{rain}$ values that differ by several orders of magnitude, in particular for submicron particles (Laakso *et al.*, 2003). It is clear, when we examine equations (1) and (2), that each of the two methods has advantages and significant limitations, which are also highlighted by the authors. The main limitation for measurement of the scavenging coefficient in the environment remains the assumption that the change in concentration is exclusively related to collection by the drops. Even if the rainfall events are methodically selected, it is difficult to completely neglect advection, turbulent transport, coagulation and the influence of the hygroscopic behaviour of particle (Flossmann, 1991). For example, Quérel *et al.* (2014a) have recently shown that during convective episodes, downdraft was the main cause of the reduction in particulate concentration, well before collection by the drops.

For the theoretical approach, the main limitation is the requirement to know the collection efficiency (equation 3). This microphysical parameter is defined as the ratio between the effective collection area (in other words, the cross-sectional area inside which the particle trajectory is intercepted by the drop) and the cross-sectional area of the drop. It is equivalent to defining the ratio of the mass of particles (of a given diameter) collected by the drop over the mass of particles (of the same diameter) within the volume swept by a sphere of equivalent volume (equation 3).

$$E\left(d_{ap}, D_{drop}, RH\right) = \frac{m_{AP,collected}(d_{AP})}{m_{AP,swept}(d_{AP})} \quad (3)$$

To compute this efficiency, one has to describe and model all the processes involved in the collection of particles by falling raindrops. Several mechanisms are usually considered, which are summarised hereafter; however, a more exhaustive review can be found in the literature (Pruppacher *et al.*, 1998; Chate, 2005; Ladino *et al.,* 2013; Ardon-Dryer *et al.*, 2015). The three main mechanisms leading to this collection are Brownian motion, inertial impaction and interception. Small particles, with a radius on the order of the mean free path of the air molecules or smaller, are very sensitive to the collision of air molecules. Therefore, they shall deviate from streamlines due to Brownian motion. For large particles, with a diameter greater than 1 µm, their inertia prevents them from following the streamlines of the flow and they impact the drop on its leading edge. Aerosol particles with a diameter smaller than 1 µm and much larger than the mean free path of the air molecules follow the streamlines of the flow around the drop. They might nevertheless enter in contact with the drop when the streamlines approach the drop at a distance smaller than the radius of the aerosol particle. For particles with diameter between 0.2 µm and 1 µm, there is a minimum collection efficiency called the "Greenfield Gap" (Greenfield, 1957). For these particles, none of the three described mechanisms is efficient for collection. It is expected that phoretic forces would be the most efficient mechanisms. To be thorough, secondary mechanisms for collision are also described here. Thermophoresis and diffusiophoresis are respectively linked to thermal and water vapour gradients. The side of a particle exposed to warmer air is impacted by molecules with higher kinetic energy than molecules impacting the colder side. As a result, thermophoresis results in a force whose direction is the opposite of the thermal gradient. Similarly, particles exposed to a water vapour gradient are exposed to molecular collisions with a dissymmetric kinetic energy since water vapour molecules are lighter than air molecules. In the atmosphere, diffusiophoresis results in a force whose direction is the opposite of the water vapour gradient. Electro-scavenging could also have an important contribution when both droplets and aerosols particles are electrically charged, resulting in an attractive (or repulsive) force when they have opposite (or identical) polarity. Moreover, Tinsley *et al.* (2000, 2006) theoretically showed that electrically charged aerosol particles can induce an image charge on droplets that results in a short range electrical attraction that increases collection efficiency even with neutrally charged droplets.

For each of these elementary mechanisms, theoretical expressions of the elementary collection efficiencies have been derived (Table 1).

Table 1. References of theoretical expressions for the calculation of each collection mechanism

| Elementary mechanism | Reference |
|---|---|
| Inertial impaction | Slinn (1977); Park *et al.* (2005) |
| Interception | Slinn (1977); Park *et al.* (2005) |
| Brownian motion | Slinn (1977); Park *et al.* (2005) |
| Diffusiophoresis | Waldmann (1959); Davenport and Peters (1978); Andronache *et al.* (2006);  Wang *et al.* (2010) |
| Thermophoresis | Davenport and Peters (1978); Andronache *et al.* (2006); Wang *et al.* (2010) |
| Electro-scavenging | Davenport and Peters (1978); Andronache *et al.* (2006); Wang *et al.* (2010) |
| Image forces | Tinsley and Zhou (2015) |

Finally, the droplet total collection efficiency can be theoretically deduced by adding all these elementary collection efficiencies together. The use of these theoretical models seems justified for cloud droplets since they have very small Reynolds numbers. However, for raindrops with larger sizes and Reynolds numbers, there are many additional uncertainties. This is because, once they reach their terminal velocity, the Reynolds and Weber numbers of these large drops are very high. They thus oscillate at high frequency (Szakáll *et al.*, 2010), which greatly complicates the simulation of flows inside and outside the drop. Furthermore, the boundary layer separation in the wake of the drop results in significant recirculating flows. Therefore, there are currently few methods for numerically simulating such flows (although the work of Menard *et al.,* 2007, shall be mentioned). The most common approach continues to be to use the Slinn model (Volken & Schumann, 1993; Laakso *et al.,* 2003; Chate, 2005; Depuydt, 2013), essentially for its ease of use and despite its simplifying assumptions. It should be kept in mind that W.G.N. Slinn models the flow around the drop as a Stokes flow, which results in ignoring the convective terms of the Navier-Stokes equation. Such flows have a similar kinematic field to that of a potential flow. The Slinn model cannot therefore capture the separation of the boundary layer in the wake of the drop. The flow on the front side of the drop is, however, relatively well modelled.

Beard and Grover (1974) have developed a numerical model that is more sophisticated than that of Slinn (1977) to numerically simulate the collision between particles and a drop. The main difference is that they do not assume Stokes flow. Flow around the drop is computed by solving the full Navier-Stokes equation including the convective term. However, Beard and Grover (1974) made two simplifying assumptions: the drop is assumed spherical and the flow axisymmetric. These simulations capture the separation of the boundary layer in the wake of the drop and the resulting recirculating flows. Using these simulations, Beard (1974) derived the collision efficiencies between drops and particles of different sizes. For this, he computed the particle trajectory in the flow considering drag and gravity forces. For the drag force, they followed the Stokes-Cunningham expression that takes into account non-continuum effects, which are important for the smallest particles. These simulations highlight, for the first time, the capture of submicron-sized particles in the rear of the drop, due to wake recirculations.

Until recently, no measurements in the numerous experimental studies (Kerker and Hampl, 1974; Grover *et al.,* 1977; Wang and Pruppacher, 1977a; Lai *et al.,* 1978; Pranesha and Kamra, 1996; Vohl *et al.,* 1999) could be used to validate these two models since very few use submicron particles. Quérel *et al.* (2014a) showed that for their dataset, the Slinn model underestimates by two orders of magnitude the measured collection efficiencies for particles with submicron sizes. As stated in the introduction, if they concluded that their data could confirm the Beard model, they were required to extrapolate the simulations of K.V. Beard to confront their observations.

In this paper, the collection efficiency is investigated experimentally for drops within the size range simulated by Beard (1974) to address these uncertainties in collection efficiency of raindrops with large Reynolds numbers by accurately measuring them in the laboratory with the ultimate aim of theoretically deriving a scavenging coefficient.

## 2    Experimental facility

The new experimental facility follows the one described and deployed by Quérel *et al.* (2014b). The equipment is called BERGAME (French acronym for a facility to study the aerosol scavenging and measure collection efficiency).

Presented in details in the following subsections, the three stages are (Figure 1):

- a mono-dispersed drop generator;
- a free-fall shaft;
- an aerosol chamber.

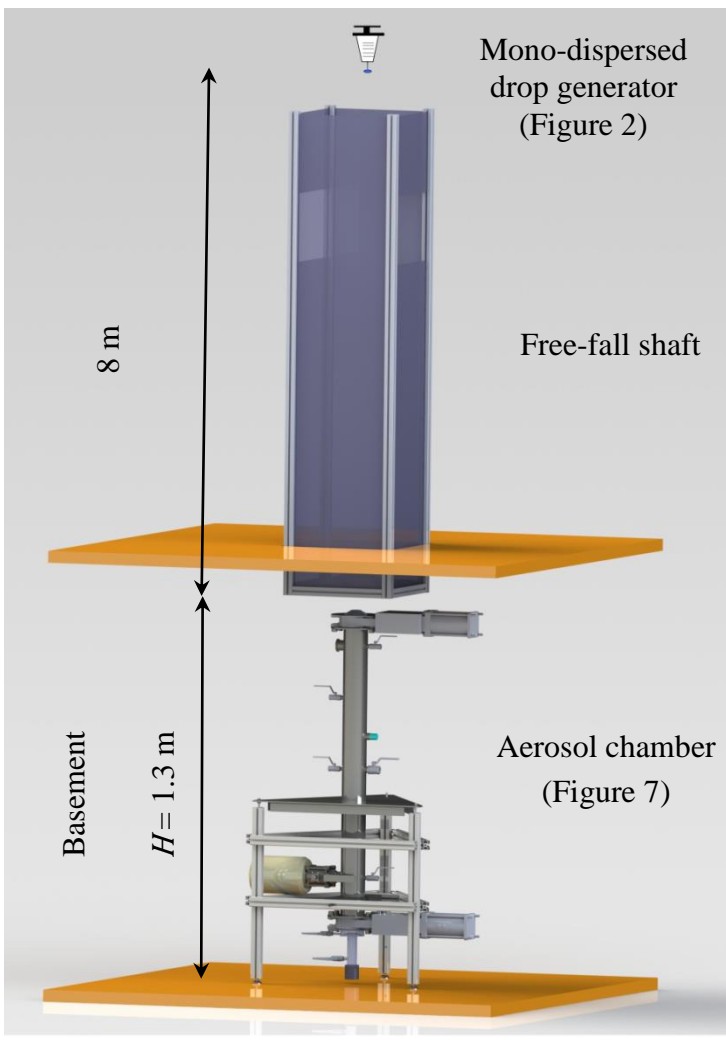

Figure 1. The new BERGAME facility

The main changes with respect to Quérel *et al.* (2014b) concern the drop generator and the aerosol chamber. Indeed, those authors concluded that drop generation has to be improved if direct comparisons with the Beard

(1974) results were to be made. Improvements are presented in subsection 2.1. In addition, the aerosol chamber has been modified not only to increase the particle number concentration, but also to better control relative humidity, to neutralise the aerosol particles, and to minimize uncertainties. The objective of these modifications is also to be consistent with the hypothesis of the Beard (1974) model, which considers only drag and gravitational forces on the aerosol particles. The modifications are thus intended to minimise electro-scavenging (discussed in sections 2.1 and 2.3), diffusiophoresis (discussed in section 2.3 and Appendix 1) and thermophoresis. Both the drop generator and aerosol chamber are described in the following sections.

## 2.1 Production of drops representative of rain

In order to enable the generation of finer drops, a new generator (Figure 2) was developed, characterised and installed at the top of the free-fall shaft of the BERGAME facility. The generator was placed 8 metres above the new aerosol chamber. The total height of the drop shaft has been reduced by 2 m because, as the drops are smaller than those investigated by Quérel *et al.* (2014a and b), they reach their terminal velocity in a shorter distance.

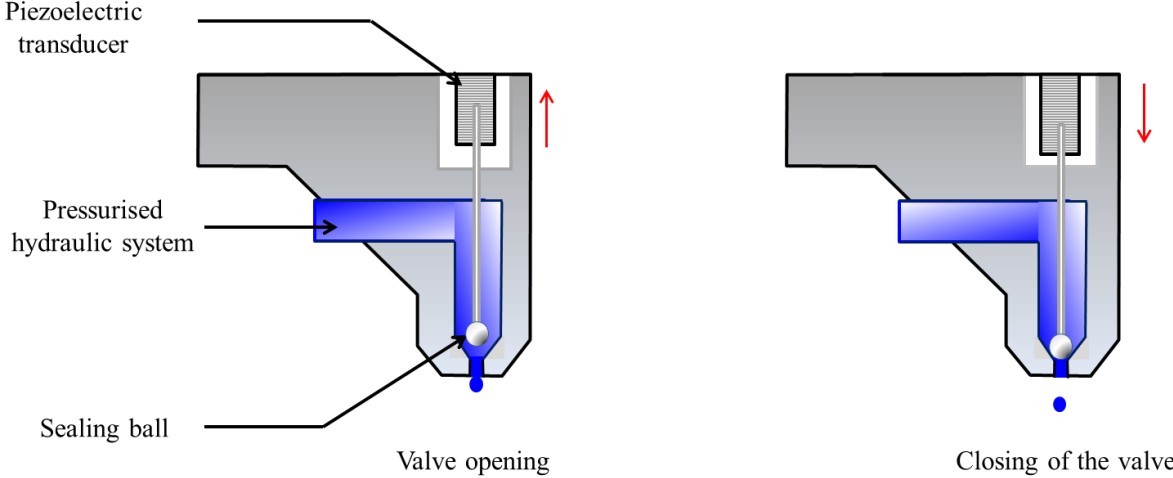

Figure 2. Diagram of operation of the generator opening valve

The drop generator consists of a valve operated by piezoelectric actuators which transmit their movement to a rod. A ceramic sealing ball is attached to the rod and lifts to open the valve by enabling the fluid to flow (see Figure 2). The water circuit is maintained under pressure by a compressed air system.

Classical piezoelectric drop-on-demand systems may produce electrically charged droplets (Ardon-Dryer *et al.,* 2015). However, we want to limit electro-scavenging as Beard (1974) did in his simulations. To control electro-scavenging, the net charge of each drop produced by this system has been measured with the help of a Faraday pail connected to an electrometer (Keithley model 6514; Sow & Lemaitre, 2016). Any electrical charge on the drop was detected by our sensitive electrometer (limit of 10 fC). This might be explained by the fact that unlike classical piezoelectric drop-on-demand systems (such as those of microdrop Technologies and MicroFab

Technologies), the piezoelectric transducer in our drop generator is not in direct contact with the liquid (Figure 2).

### *Drop size measurements*

The generator was calibrated in order to produce drops of a prescribed diameter. Two parameters govern the size of the drops: the water supply pressure and the valve opening time. The different tests performed showed that when the pressure in the water circuit is too high, the drops break up at the injector outlet. Maintaining pressure below, or at 0.3 bar, avoids these effects. These tests were therefore performed at a positive pressure of 0.3 bar. For this water circuit supply pressure, the valve opening time was between 4 and 11 ms. The raindrops size is determined after a free-fall acceleration over a height of 8 m. For each opening time, shadowgraph measurements were taken in the aerosol chamber of the BERGAME facility. An optical window is used to trigger the photographing of each drop entering the BERGAME aerosol chamber. Our optical device is a camera (Andor: neo, sCMOS) with a resolution of $2560 \times 2160$ pixels². It is equipped with a Canon macro lens (MP-E 65mm f/2.8 1-5x) for a magnification of 3:1 (experimentally checked with a calibration chart). The pixel size is 6.5 µm, for a spatial resolution of 2.1 µm. Drops are backlighted with a 9 ns strobe to freeze their fall on the sensor. An example of a shadowgraph image is shown in Figure 3.

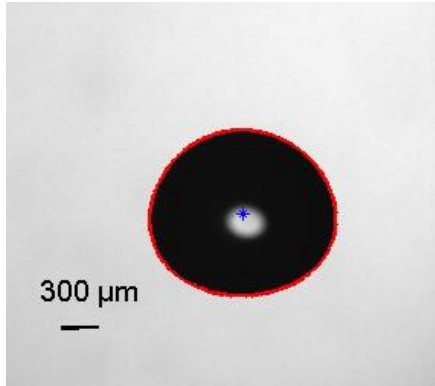

Figure 3. Example of a shadow image

Due to the oscillations, the millimetric drops exhibit an oblate spheroid shape. To define the size of the raindrops the notion of "diameter equivalent to a sphere of the same volume" has been adopted. Since shadowgraphy yields only a 2-D information, the diameters are equivalent to a disc. For axisymmetric objects, volume and surface equivalent diameter are equal. Szakáll *et al.* (2009) experimentally verified this axisymmetric of drop of that size range at terminal velocity. Thus, shadow images are used and processed to deduce the projected surface area of the drop ($S_{drop}$) and derive the diameter of the disc of equal surface area ($D_{eq}$).

$$D_{eq} = \sqrt{\frac{4\,S_{drop}}{\pi}} \qquad (4)$$

For each injection configuration, the equivalent diameter of the drops is measured for one hundred images. Finally, the mean equivalent diameter and the standard deviation are calculated. Figure 4 shows all the measurement points investigated. For all operating points, the standard deviation is approximately 20 µm, i.e., approximately 1.5% of the size of the drop.

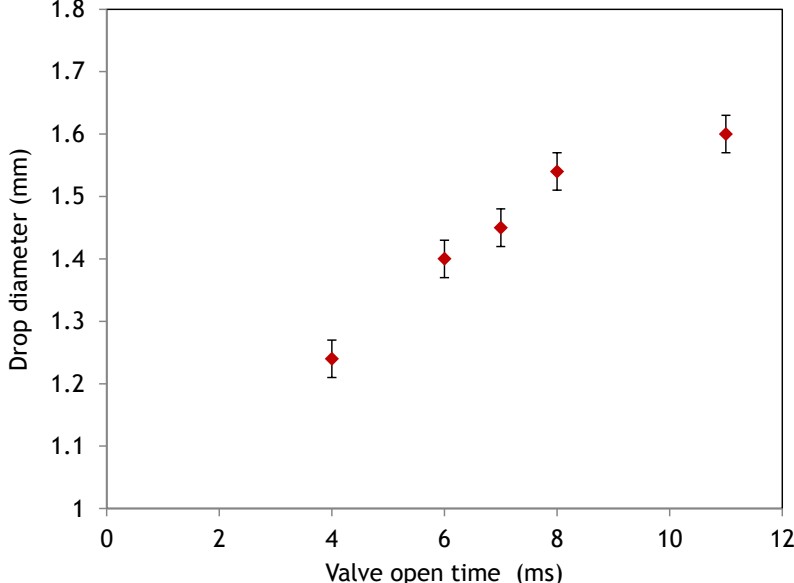

Figure 4. Measured equivalent diameter of the drop produced by our generator as a function of the valve opening time (for an over pressure of 0.3 bar)

### *Drop velocity measurements*

In order to be representative of rain the drops must cross the BERGAME aerosol chamber at their terminal velocity. For each of the drop sizes produced by our generator, the drop fall velocity is also measured at the entrance of the aerosol chamber, below the 8 m free fall shaft. Two consecutive pictures of the same drop are

10   taken during its fall. By knowing the time interval between these two images and measuring the displacement of the centre of the drop, we derive its velocity. The results are shown in Figure 5 and compared to the theoretical values computed from Beard (1976), often taken as the reference in the literature as it was validated both in wind tunnel tests and in the environment.

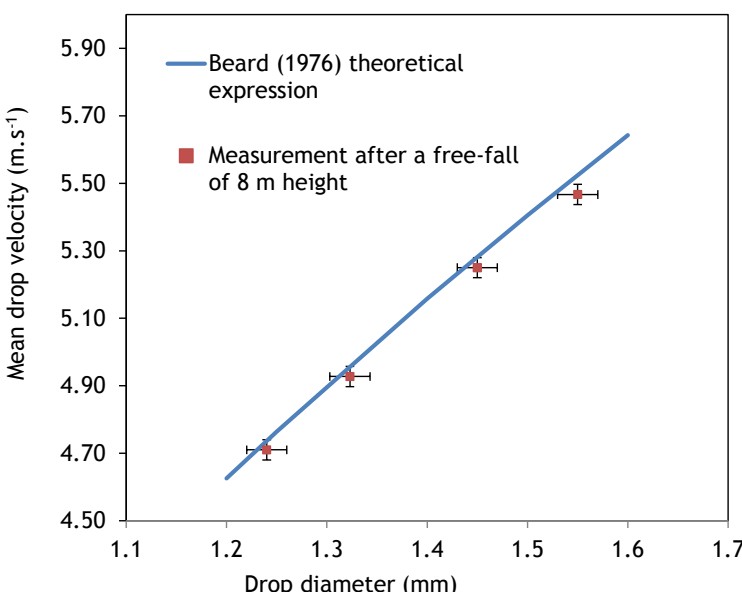

15          Figure 5. Comparison of velocities measured in BERGAME with the Beard (1976) model

We note in this figure that up to a drop diameter of 1.4 mm, the 8 m distance is sufficient to accelerate the drops to their terminal velocity. This is consistent with the results of the theoretical calculations of Wang and Pruppacher (1977b), which predict that 6.5 m free fall is enough for a 1.4 mm drop to reach 99% of terminal velocity. Furthermore, to ensure that our drops are representative of the hydrometeors described in the literature, we compare in Figure 6 the axis ratios of the drops in the BERGAME chamber with the model of Beard and Chuang (1987). For the drop sizes investigated, drop can be considered as horizontally aligned oblate spheroids (Figure 3), no tilt angle was measured, which is consistent with Pruppacher & Beard (1970) measurements. This is why, the axis ratio is computed as the ratio between the vertical and horizontal dimensions of the drop.

Figure 6 shows that up to a diameter of 1.4 mm the drops entering the aerosol chamber are perfectly representative of the hydrometeors observed in the atmosphere.

In this study, we focus on the collection efficiency of drops with a diameter of 1.25 mm. We have selected this size, because it is the only one produced by our systems for which comparisons with Beard (1974) simulations will be direct. This model is particularly interesting as we have previously shown that for 2 mm diameter drops (Quérel *et al.,* 2014b), it is the only one able to predict the sharp rise in the collection efficiency observed experimentally for sub-microscopic particles, which is due to the eddies that develop within the wake of the drop. These vortices will capture the particles and draw them back onto the rear of the drop. For a drop diameter of 1.25 mm, an 8 m free fall distance is enough for the drops to represent atmospheric raindrops, both in terms of velocity and axis ratio.

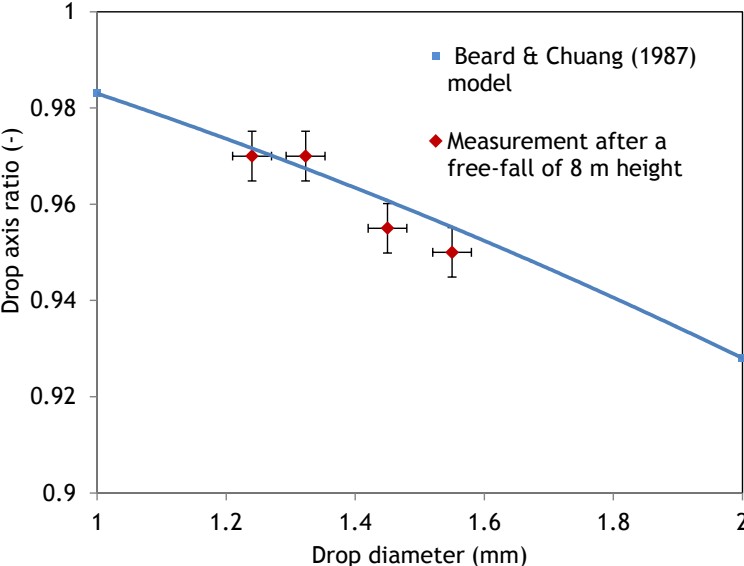

Figure 6. Comparison of axis ratios measured in BERGAME with the model of Beard and Chuang (1987)

## 2.2   Description of the new BERGAME aerosol chamber

A new aerosol chamber (Figure 7) has been designed to increase the concentration of particles within the volume swept by the drops during their fall. Its geometry is strongly influenced by the one developed by Hampl *et al.* (1971). It consists of a 1,300 mm high stainless steel cylinder with an internal diameter of 100 mm.

Various taps are provided for injecting the aerosols, taking samples and characterising the thermodynamic conditions of the gas. These various sampling points serve to measure in particular:

- ⁃   the aerosol particle size distribution,
- ⁃   their mass concentration,
- ⁃   the temperature and relative humidity.

In Figure 7, each valve is labelled with a Greek letter to structure the explanations in the text. The chamber is fitted with two gate valves, one at the top ($\kappa$) and the other at the bottom ($\varphi$). These two valves isolate the chamber while it is being filled with particles. The particle size distribution of the aerosols is measured by means of an Aerodynamic Particle Sizer (APS, $\chi$) and an Electrical Low Pressure Impactor (ELPI, $\delta$). The injected particles are pure fluorescein particles so that they may be easily measured by fluorescence spectrometry. The mass concentration of the particles in suspension inside the chamber is determined by venting the entire content of the chamber onto a High Efficiency Particulate Arresting (HEPA) filter ($\alpha$), and measuring the mass of particles on the filter using fluorescence spectrometry.

Finally, the relative humidity and the temperature are given respectively by a capacitive hygrometer and a thermocouple ($\omega$).

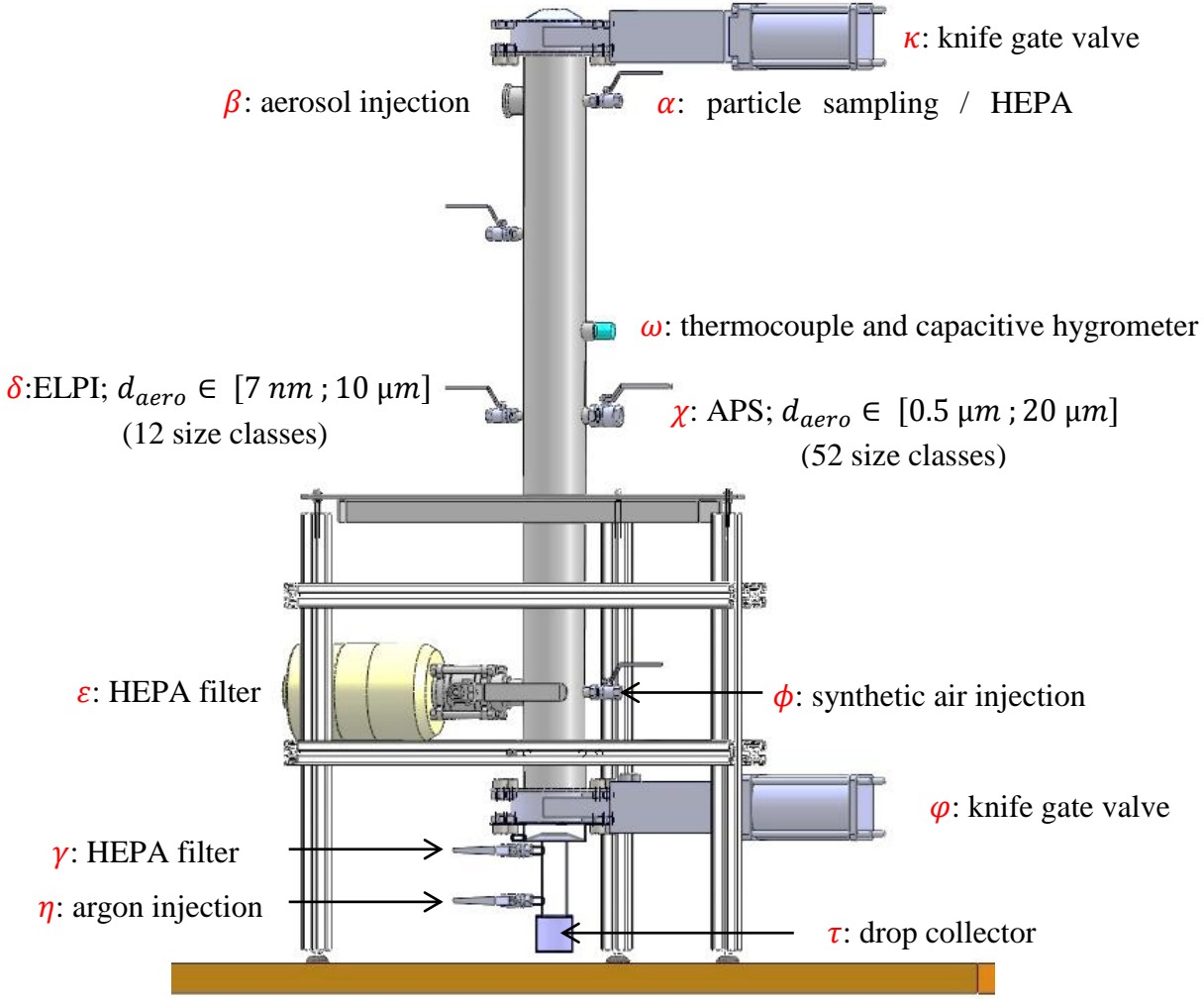

$\kappa$: knife gate valve

$\beta$: aerosol injection $\qquad$ $\alpha$: particle sampling / HEPA

$\omega$: thermocouple and capacitive hygrometer

$\delta$:ELPI; $d_{aero} \in [7\ nm\ ;10\ \mu m]$ $\qquad$ $\chi$: APS; $d_{aero} \in [0.5\ \mu m\ ;20\ \mu m]$
(12 size classes) $\qquad\qquad\qquad$ (52 size classes)

$\varepsilon$: HEPA filter $\qquad\qquad\qquad\qquad$ $\phi$: synthetic air injection

$\qquad\qquad\qquad\qquad\qquad\qquad\qquad\qquad$ $\varphi$: knife gate valve

$\gamma$: HEPA filter

$\eta$: argon injection $\qquad\qquad\qquad\qquad\qquad$ $\tau$: drop collector

Figure 7. Schematic design of the new BERGAME aerosol chamber

After having accelerated in free fall over 8 m, the drops are representative of rain in terms of size, velocity and axis ratio (section 2.1). They enter the aerosol chamber, via a circular opening with a four centimetres diameter. After crossing the aerosol chamber, the drops are collected in a removable container ($\tau$). One of the principal difficulties of these experiments relates to the sedimentation of the cloud of particles that settles directly inside the drop collector. Indeed, Rayleigh-Taylor instabilities can arise when a dense cloud of aerosol particles overlies

a layer of clean air. These instabilities induce a downward motion of the aerosol cloud much faster that the settling velocity of individual particles (Hinds *et al.,* 2002). In order to avoid this effect, a layer of argon (which is denser than the cloud of particles) is formed in the bottom of the aerosol chamber, located below the second gate valve in Figure 7. A large number of experiments were performed. These experiments show that, regardless of the concentration and the size of the particles in the aerosol chamber, until four minutes after opening the gate

valves, the drop collector is free from any particulate contamination. Beyond four minutes, traces of fluorescein are detected on the drop collector.

### 2.3 Aerosol particle characterisation and generation

The aerosol particles size distributions are measured using an Electrical Low Pressure Impactor (ELPI, $\delta$) and an Aerodynamic Particle Sizer (APS, $\chi$).

ELPI is a quasi-real-time aerosol spectrometer (Marjamäki *et al.*, 2000). It is composed of a corona charger and a 12-stage cascade low pressure impactor. Each stage of the impactor is connected to an electrometer. The corona charger is used to set the electrical charge of the particles to a specific level. Then, the low pressure impactor classifies the aerosol particles into 12 size classes according to their aerodynamic diameter (from 7 nm to 10 μm). Finally, the electrometers measure the electrical charge carried by the particles collected by each impaction stage. This charge is finally converted to the number of particles collected according to the charging efficiency function of the corona charger.

APS is also a quasi-real-time aerosol spectrometer (Baron, 1986). It measures the time-of-flight of individual particles accelerated by a controlled accelerating flow imposed by a calibrated nozzle. The time-of-flight of each aerosol particle is then converted into its aerodynamic diameter. Thus, the APS classifies the aerosol particles in terms of aerodynamic diameter from 500 nm to 20 μm over 52 size classes.

APS and ELPI are both used for their complementary size ranges so all the particles produced in our laboratory can be sized. For particles with a median aerodynamic diameter less than 0.8 μm, the size distribution is measured using an ELPI. For the others, we favour the use of an APS because of the better size resolution.

The aerosol particles are produced with two ultrasound generators. The key part of these generators is a piezoelectric ceramic immersed in a solution. When subjected to an appropriate electric current, this ceramic vibrates at a frequency of 500 or 2 400 kHz depending on the generator used.

These oscillations transform the surface of the liquid into a mist of microscopic droplets with a narrow size distribution. These drops are transported to the upper part of the generator, by a flow of dry filtered air at a flow rate of 20 L.min$^{-1}$. More dry air is added in the upper part of the generator at a flow rate of 30 L.min$^{-1}$ to dry the particles.

These drying and dispersal flow rates have been selected to obtain the following characteristics:

- the aerosol particle size distributions are narrowly spread (geometric standard deviation less than or equal to 1.5),
- the particle concentration inside the aerosol chamber is high ($\sim 2 \times 10^5$ particles.cm$^{-3}$),
- the relative humidity measured in the aerosol chamber is approximately $77 \pm 1\%$. This humidity corresponds to relative humidities observed during rainfall events (Depuydt *et al.,* 2012). Furthermore, we will show that this humidity is high enough to make diffusiophoresis negligible (see the discussion of Figure 12, section 3).

Changing the concentration of the solute dissolved in the water varies the size of the produced particles. The chosen solute is sodium fluorescein ($C_{10}H_{10}Na_2O_5$). This molecule has been selected for its very large fluorescence properties. It can be easily detected by fluorescence spectroscopy down to a concentration of $5 \times 10^{-11}$ g.mL$^{-1}$. The generator is placed inside a negative pressure enclosure to prevent any possible fluorescein

particle contamination of the laboratory. Figure 8 shows two examples of number particle size distributions of fluorescein measured in the BERGAME aerosol chamber.

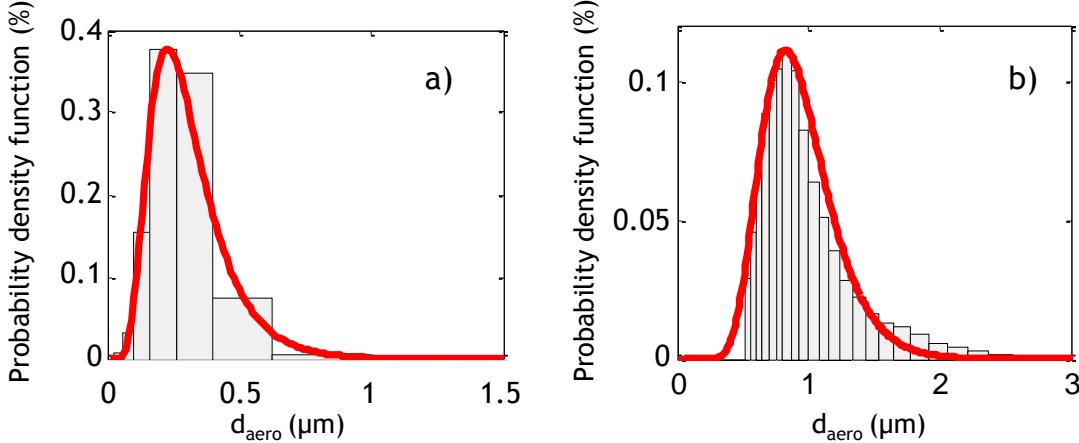

Figure 8. The size distribution of the particles produced by the ultrasound aerosol generator vibrating at 2400 kHz: a) for a fluorescein concentration of 0.11 g.L$^{-1}$ b) for a fluorescein concentration of 10 g.L$^{-1}$. The distribution on the left is measured using an electrical low pressure impactor (ELPI, $\delta$) and the one on the right using an aerodynamic particle sizer (APS, $\chi$).

Both of these distributions fit well to log-normal distributions (red curves on the graphs). For a fluorescein concentration of 0.11 g.L$^{-1}$ (respectively 10 g.L$^{-1}$) in the solution, the median diameter of the fitted distribution is 220 nm (respectively 820 nm) and the geometric standard deviation is 1.5 (respectively 1.34).

For each of the particle sizes produced, the fluorescein mass concentrations in the aerosol chamber derived from APS and ELPI measurements are compared with ones derived from filter measurements (section 2.2). These comparisons provide slight differences (∼10%) that can be attributed to both the purity of fluorescein sodium salt used (∼97%) and the shape of the aerosol particles that is not perfectly spherical. Thus, for improving the accuracy of collection efficiency measurements, the fluorescein concentration inside the aerosol chamber is derived from filter measurements, and APS and ELPI are used to provide a precise measurement of the particle size.

In order to neutralise the charge of the aerosol particles prior to injecting them into the BERGAME aerosol chamber ($\beta$), the particles go through a low energy X-ray neutraliser (< 9.5 keV, TSI 3088), at a flow rate of 1.5 L.min$^{-1}$. At this flow rate, the residence time of the particles in the neutraliser is sufficient to neutralise them.

As we have seen in the previous section, our aerosol generator produces aerosols at a flow rate of 50 L.min$^{-1}$ (20 L.min$^{-1}$ of dispersion air and 30 L.min$^{-1}$ of drying air). We use, therefore, a flow divider to ensure that the particles pass through the neutraliser at 1.5 L.min$^{-1}$. This divider includes an 8 litre buffer volume, provided with one inlet and two outlets. A flow rate of 48.5 L.min$^{-1}$ is drawn-off from one of these outlets by means of an air suction pump. This flow is filtered and vented. The remaining flow passes through the neutraliser. After neutralisation, the particles are injected into the aerosol chamber.

### 2.4 Test procedure

The aerosol chamber is flushed at the start of each experiment with synthetic air to ensure that initial conditions are free of any fluorescein particle contamination. After flushing, the previously neutralised aerosol particles of chosen diameter are injected at a flow rate of 1.5 L.min$^{-1}$ via valve $\beta$ (section 2.3).

The two knife gate valves ($\varphi$ and $\kappa$) are closed during this filling phase in order to isolate the enclosure. In addition, valve $\varepsilon$ is opened to exhaust the excess pressure towards a HEPA filter. The injection process lasts 20 minutes, during which we form a layer of argon within the zone located below knife gate valve φ. This injection is carried out in two stages. First, we inject the argon during 10 minutes via valve η, with the drop collector unscrewed and valve $\gamma$ closed. Second, the drop collector is refitted and valve $\gamma$ is opened. At the end of this phase, the aerosol chamber is filled with neutralised particles of a prescribed diameter at a concentration of approximately $2 \times 10^5$ particles per cubic centimetre.

This filling phase of the enclosure is followed by a relaxation period lasting no less than 15 minutes. During this time period, all the valves of the aerosol chamber are closed with the exception of valve ε, which remains open in order to perfectly balance the pressures. This period is used to bring the train of drops produced by the generator at the centreline of the aerosol chamber. Once the drop generator is adjusted, valve $\varepsilon$ is closed and both knife gate valves ($\varphi$ and $\kappa$) are opened to enable the drops to cross the aerosol chamber. A cumulated volume of 1 cm$^3$ of solution is necessary for performing a measurement by fluorescence spectrometry, i.e., approximately 1000 drops of 1.25 mm diameter. As a result of the frequency at which drops cross the enclosure, 10 minutes are needed to collect this volume. As mentioned above, the drop collector remains free of any particulate contamination if the valves remain open for less than 4 minutes. The 10 minutes needed to collect the 1000 drops are therefore divided into 3 periods of 200 seconds each. At the end of these 200 seconds phases, the gate valves are closed again and the buffer volume between gate valve φ and the drop collector is flushed with argon (Figure 9). During flushing, the argon is injected through valve $\eta$ and removed through valve $\gamma$, which ensures an upward flow within this buffer volume and minimises the risk of contamination of the drop collector.

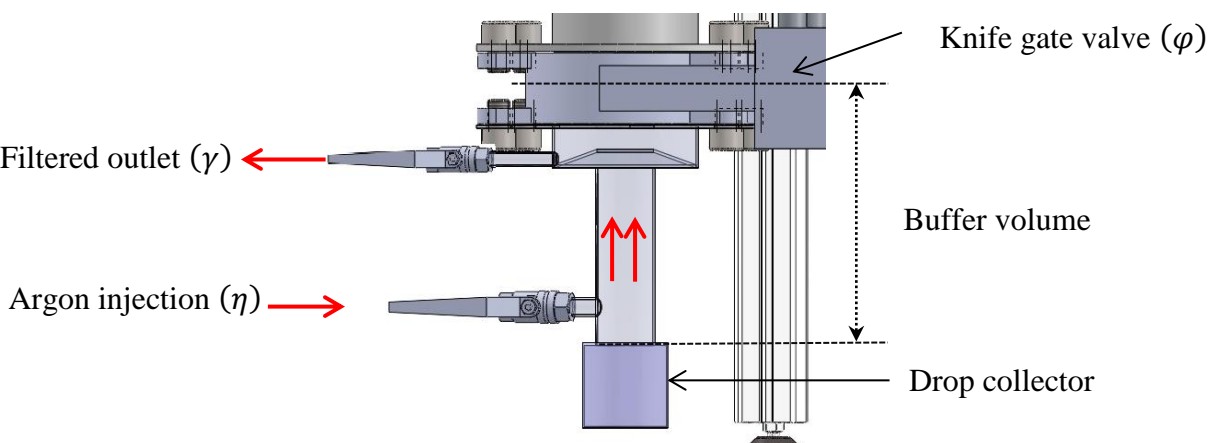

Figure 9. Buffer volume flushing procedure

Once 1 cm$^3$ of drops collected, both knife gate valves close, and the buffer volume is flushed, to avoid any contamination of the collected water when removing the drop collector.

In order to determine the collection efficiency, we need to know the mass concentration of fluorescein within the volume swept by the drops (eq. 3). The concentration is measured by filter analysis and in this purpose the aerosol chamber of the BERGAME experiment is flushed with synthetic air at the end of each experiment. This is done by injecting the synthetic air through valve $\phi$, at a flow rate of 5 L.min$^{-1}$ during 10 minutes, and collecting the particles on a HEPA filter.

This filter is then placed in 100 mL of ammonia water ($V_{sol}$) for 24 hours in order to dissolve all the fluorescein particles it contains. Finally, the mass concentration of fluorescein in this water ($[fluo]_{filter}$) is measured by fluorescence spectrometry.

The mass concentration of fluorescein particles in the aerosol chamber ($[fluo]_{chamber}$) is then determined using the following equation:

$$[fluo]_{chamber} = \frac{[fluo]_{filter} \cdot V_{sol}}{V_{chamber}} \quad . \quad (5)$$

In this equation, the term $V_{chamber}$ is the volume of the aerosol chamber, i.e. 10.2 L.

As the mass concentration of particles is only quantified once the measurements are completed, we have attempted to quantify its variation over the duration of a measurement (approximately 15 min). For this, we have first verified the reproducibility of characteristics of the aerosol produced by the aerosol generator, in size, number and concentration. This is performed by repeating the injection phase with exactly the same operating conditions. No variation of the fluorescein concentration greater than the uncertainty of the fluorimeter ($\pm 2.5\%$, appendix 1) has ever been measured. We have then compared the mass concentration in the aerosol chamber just after the relaxation phase and after a complete measurement procedure. At last, we measured a reduction in concentration of less than 8% regardless of the particle diameter. These particles are essentially lost through deposition on the sides of the aerosol chamber. This decrease of the particle concentration during the experiments is the main source of uncertainty on the measurement of the collection efficiency.

The collection efficiency is defined as the ratio between the mass of particles (of a given diameter) collected by a drop as it falls, and the total mass of particles (of the same diameter) within the volume it has swept. The mass of fluorescein in the drops during the experiments ($M_{drop}$) is easy to calculate:

$$M_{drop} = \frac{\pi D_{drop}^3}{6} [fluo]_{drop} \quad (6)$$

where $[fluo]_{drop}$ is the mass concentration of fluorescein in the drops.

The mass of particles within the volume swept by the drops ($M_2$) is calculated with:

$$M_2 = \frac{\pi D_{drop}^2 H}{4} [fluo]_{chamber} \quad (7)$$

where $[fluo]_{chamber}$ is the mass concentration of fluorescein in the aerosol chamber and $H$ the height of the aerosol chamber (1.3 m, Figure 1).

The collection efficiency is derived from the following expression:

$$E\left(d_{aero}, D_{drop}, RH\right) = \frac{M_{drop}}{M_2} = \frac{2\,D_{drop}.[fluo]_{drop}}{3H.[fluo]_{chamber}} \quad (8)$$

In order to precisely determine the size distribution of the particles for which the collection efficiency has been measured, we repeat the injection of particles into the BERGAME aerosol chamber following each efficiency measurement under exactly the same operating conditions (generator, ceramic excitation frequencies, injection times, dispersal and drying flow rates and fluorescein concentration). The size distribution of the aerosol particles produced by the generator is then measured in the aerosol chamber.

## 3   Results and Discussion

All the measurements taken are summarised in Table 2 with the associated expanded relative uncertainties. The first column of this table provides the median aerodynamic diameter ($d_{aero}$) of each particle size distribution measured using the APS or the ELPI. The detailed calculation of the uncertainties is presented in Appendix 1 (Lira, 2002).

Table 2. Summary of measurements performed

| $d_{aero}$ (µm) | $D_{drop}$ (mm) | RH (%) | $[fluo]_{drop}$ (g.cm$^{-3}$) | | $[fluo]_{chamber}$ (g.cm$^{-3}$) | | $E$ (-) | |
|---|---|---|---|---|---|---|---|---|
| 0.25 | | | $8.22 \times 10^{-8}$ | | $6.22 \times 10^{-9}$ | | $8.8 \times 10^{-3}$ | |
| 0.25 | | | $1.15 \times 10^{-7}$ | | $7.91 \times 10^{-9}$ | | $9.7 \times 10^{-3}$ | |
| 0.5 | | | $3.39 \times 10^{-8}$ | | $4.22 \times 10^{-9}$ | | $5.4 \times 10^{-3}$ | |
| 0.6 | | | $4.51 \times 10^{-8}$ | | $1.38 \times 10^{-8}$ | | $2.2 \times 10^{-3}$ | |
| 0.71 | $1.25 \pm 1.5\%$ | $77 \pm 5\%$ | $2.15 \times 10^{-8}$ | $\pm 2.5\%$ | $9.62 \times 10^{-9}$ | $\pm 8\%$ | $1.5 \times 10^{-3}$ | $\pm 16\%$ |
| 1 | | | $2.52 \times 10^{-7}$ | | $1.17 \times 10^{-9}$ | | $2.9 \times 10^{-3}$ | |
| 1.47 | | | $5.48 \times 10^{-8}$ | | $6.39 \times 10^{-9}$ | | $5.7 \times 10^{-3}$ | |
| 2.54 | | | $6.51 \times 10^{-7}$ | | $5.51 \times 10^{-9}$ | | $7.9 \times 10^{-2}$ | |

This median aerodynamic diameter is converted into a physical diameter ($d_{ap}$) by means of the following expression (which is solved iteratively):

$$d_{ap} = d_{aero} \sqrt{\frac{C_{c,d_{aero}}}{C_{c,d_{ap}}}\left(\frac{\rho_0}{\rho_p}\right)} \quad (9)$$

In this equation $C_c$ is the Cunningham slip correction factor and $\rho_0$ the water density. The density of the particle $\left(\rho_p\right)$ is calculated from the growth factor ($GF$) of the fluorescein aerosol particle.

$$\rho_p = \frac{\rho_{C_{10}H_{10}Na_2O_5} + \rho_0(GF^3 - 1)}{GF^3} \quad (10)$$

This factor has previously been measured using a Hygroscopicity Tandem Differential Mobility Analyser (HTDMA, Quérel *et al.,* 2014b). For our experiments, performed at a relative humidity of 77 ±5%, we deduce a growth factor (GF) of 1.25 ±0.05. Stöber and Flachsbart (1973) have measured a density of 1.58 g.cm[-3] for a dry fluorescein aerosol particle. Using equation 10, we therefore calculate the density of our aerosol in the aerosol chamber to be $1.30 \pm 0.05$ g.cm[-3].

The aerodynamic diameters measured in the aerosol chamber by the APS and the ELPI can then be expressed as physical diameters ($d_{ap}$):

$$d_{ap} = 0.88 \times d_{aero} \quad (11)$$

All our measurements are summarised in Table 2 and plotted in Figure 10 as a function of the median diameter of the distribution of the physical diameter of the particles. Figure 10 compares our dataset against the efficiencies computed by both Slinn (1977) and Beard (1974) models. In this figure, the Slinn model includes the contributions of inertial impaction, Brownian diffusion and interception (Table 1). It should be remembered that the *in situ* scavenging measurements (Volken and Shumann, 1993; Laakso *et al.*, 2003; Chate, 2005) are only compared to the Slinn model. For aerosol particles with diameter in the accumulation mode, the measured collection efficiencies vary considerably with the particle size. On a logarithmic scale, the efficiency curve obtained has a "V" shape with a minimum around 0.65 µm. The increase in collection efficiency for particles larger than 0.65 µm is attributed to the mechanism of impaction on the front side of the drop. Within this size range, the increase in the diameter of the particle increases its inertia. The particle can then no longer follow the streamlines and impacts the drop.

The reasons for the increase in collection efficiency for particles smaller than 0.65 µm in diameter are not as easy to figure out. Indeed particles of this size range are not expected to be affected by Brownian motion since their diameter is seven times bigger than the mean free path of the air molecules.

The Slinn model does not predict this increase and underestimates the collection efficiency for a 0.22 µm particle by two orders of magnitude. This is linked to the assumptions of Stokes flow around the drop. Yet, at Reynolds numbers larger than 20 (for a 280 µm drop at its terminal velocity), recirculation eddies develop in the wake of the drop. Beard (1974) has shown the major influence of these wake vortices on the collection of submicron-sized particles. In fact, he showed that the smallest aerosol particles are trapped in these eddies in the wake of the drop and then collected on its rear side.

This model is not referred to in the literature as it has never been validated by experiment until now. Yet we observe that, for particles below this minimum efficiency, our measurements are in almost perfect agreement with the model and seem to validate it.

For particles with a diameter greater than 1 µm, we observe that the Beard or Slinn models yielded almost the same values. This result is expected since their only difference stands in the Stokes flow around the drop for Slinn model. This assumption prevents the capture of boundary layer separation in the wake of the drop and the resulting recirculating flows even if it makes very little difference to the flow on the leading edge of the drop. Yet particles with a diameter greater than 1 µm are very sensitive to inertial effects and are captured on this front

side. Moreover, as the Stokes number of these large particles is high, they pass through the recirculations without being trapped.

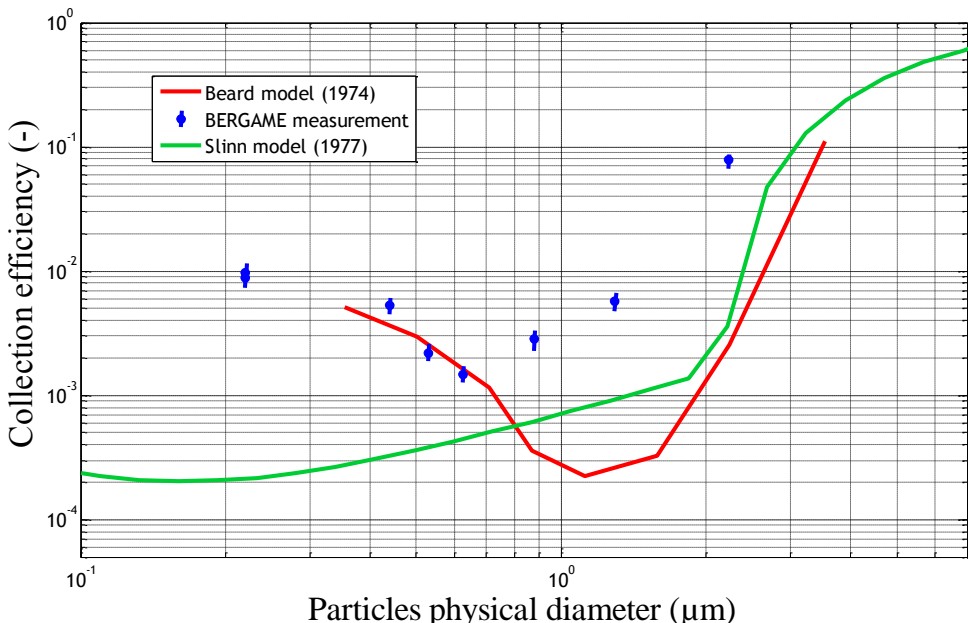

Figure 10. Comparison of our measurements for a drop of 1.25 mm diameter with the results of the models of Beard (1974) and Slinn (1977)

For particles with a diameter greater than 0.65 µm, our measurements show the same trends as these two models but with an average difference of one order of magnitude. This is probably related to the fact that, during our experiments, the aerosol particles in the aerosol chamber are not perfectly mono-disperse. Indeed, the particles have log-normal distributions with geometric standard deviations between 1.3 and 1.5 (Figure 8). The collection efficiency varies very sharply with particle size. Thus, in order to compare more rigorously our measurements with the Beard (1974) model, we need to calculate, for each measurement, the average theoretical collection efficiency $\left(\langle E\left(D_{gtte}, d_{ap}\right)\rangle\right)$ resulting from the integration of the Beard (1974) model over the entire range of particle sizes in the aerosol chamber (Eq. 12).

$$\langle E\left(D_{drop} = 1.25\ mm, d_{ap}\right)\rangle = \frac{\int_{d_{ap=0}}^{\infty} f\left(d_{ap}\right).d_{ap}^3.E\left(D_{drop} = 1.25\ mm, d_{ap}\right)\mathrm{d}d_{ap}}{\int_{d_{ap=0}}^{\infty} f\left(d_{ap}\right).d_{ap}^3.\mathrm{d}d_{ap}} \quad (12)$$

In this equation, the term $f\left(d_{ap}\right)$ is the probability density function according to the number of the particles in the BERGAME aerosol chamber, and $E(D_{drop} = 1.25\ mm,\ d_{ap})$ is the collection efficiency calculated by the Beard model (1974) for a drop 1.25 mm in diameter. The numerator and denominator of this equation are both weighted by a term $d_{ap}^3$, which reflects the fact that, experimentally, we measure intensities of fluorescence, and therefore masses of particles. We use the rectangle method to numerically solve this integral. In addition, the functions $E(D_{drop} = 1.25\ mm,\ d_{ap})$ and $f\left(d_{ap}\right)$ are both interpolated using Hermite interpolation polynomials (Fritsch and Carlson, 1980) with a step size of 0.1 µm.

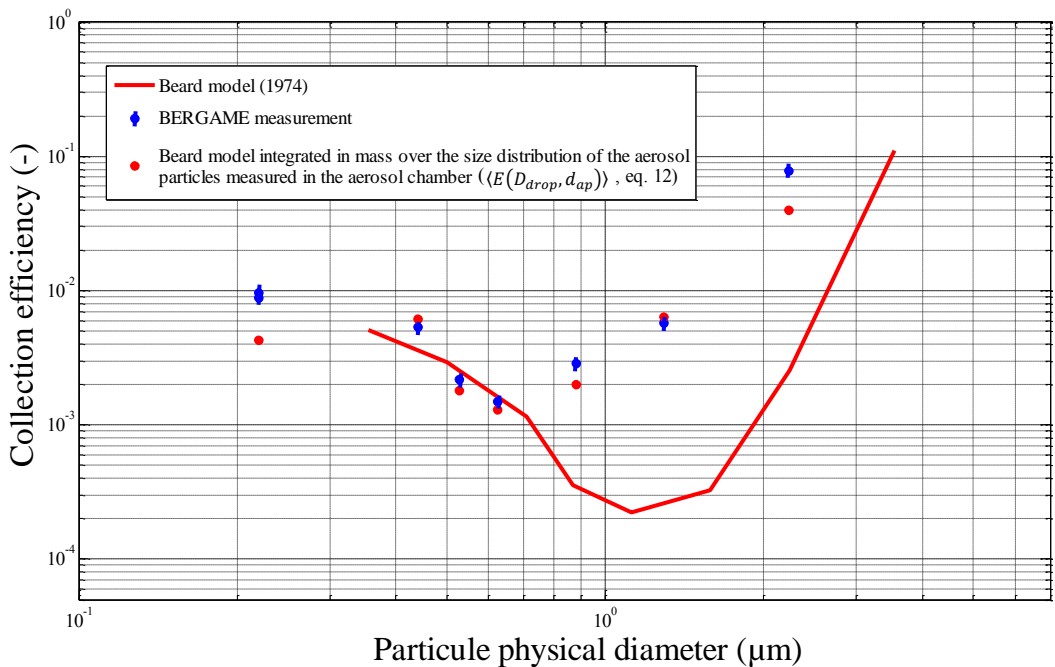

Figure 11. Integration of the Beard (1974) model over the particle size distribution of each of our experiments, for a drop of 1.25 mm diameter.

We note a significant improvement of the agreement between our measurements and the Beard (1974) model since it is integrated over the entire particle size distribution measured during our experiments in BERGAME (red dots on Figure 11). Larger differences are nevertheless observed for the first ($d_{ap} = 0.22\ \mu m$) and last measurement points ($d_{ap} = 2.54\ \mu m$). These differences could be attributed to the fact that, for these points, the resolution of equation (12) requires an extrapolation of Beard (1974) calculations beyond the size range he

investigated (continuous line on Figure 11).

Moreover, for the collection efficiency measured for the finest aerosol particles ($d_{ap} = 0.22\ \mu m$), the discrepancy observed with the Beard model could also be explained by the hypothesis of the simulations. Indeed, the Brownian motion was neglected. This can be justified in the particle size range investigated; however, it is much less justified when extrapolating the simulations to finer aerosol particles.

Furthermore, it is interesting to compare our measurements with the ones from Lai *et al.* (1978) since they are the only ones in the literature in the same drop size range. As the aerosol particles produced in these experiments are composed of silver chloride ($\rho_{AgCl} = 5.6\ g.cm^{-3}$), which is much denser than sodium fluorescein ($\rho_{C_{10}H_{10}Na_2O_5} = 1.3\ g.cm^{-3}$), it is more appropriate to plot all the collection efficiencies as a function of the Stokes number of the particle ($St_{ap}$).

$$St_{ap} = \frac{\rho_p U_\infty \left(D_{drop}\right) d_{ap}^2 C_{c,d_{ap}}}{9 D_{drop} \mu_{air}}$$

In this equation, $\mu_{air}$ is the dynamic viscosity of the air and $\rho_p$ the density of the aerosol particles. This comparison is presented on Figure 12.

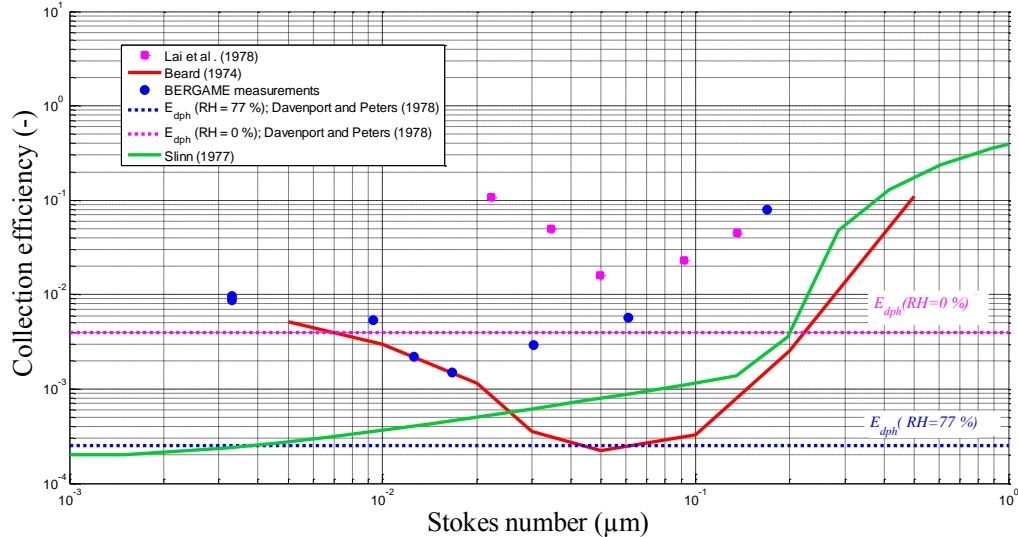

Figure 12. Collection efficiencies measured in this study and by Lai *et al*. (1978). Both measurements are compared to Slinn (1977) and Beard (1974) models. The contribution of diffusiophoresis in both studies are computed following the description of Davenport and Peters (1978)

For particles with a Stokes number greater than $6\times10^{-2}$, the motion of the particles is driven by their inertia, leading us to expect to observe the same trends in our measurement and those of Lai *et al.* (1978). The comparison for Stokes number smaller than $6\times10^{-2}$ is much less obvious. Indeed, for these particles, the measurements of Lai *et al.* (1978) indicate an increase in collection efficiency, while our measurements continue to decrease down to a Stokes number of $1.6\times10^{-2}$. At that point, the slopes of the increases of both collection efficiency measurements are similar, while the Stokes number decreases.

A precise analysis of the procedure for the aerosol particle injection in the experiments of Lai *et al.* (1978) indicates that the carrier gas is pure nitrogen without any subsequent humidification. As a consequence, it is reasonable to consider that their measurements were performed with 0% relative humidity. In order to compare the contribution of diffusiophoresis for both our experiment and that of Lai *et al.* (1978), we plot in Figure 12 the elementary contribution of diffusiophoresis ($E_{dph}$) to the collection efficiency. This contribution is calculated with the Davenport and Peters (1978) model for 0% relative humidity (as expected for the experiments of Lai *et al*., 1978) and 77% (as measured in our experiments). From this figure, it will be noted that for the experiments of Lai *et al.* (1978), the contribution of diffusiophoresis is more than one order of magnitude higher than in ours. Furthermore, while in our experiments the contribution of diffusiophoresis is smaller than the collection efficiency simulated by Beard (1974), the opposite is observed with Lai *et al.* (1978). Thus, it appears that the experiments of Lai *et al.* (1978) cannot be compared directly to Beard (1974)'s model, because they seem to be dominated by diffusiophoresis.

Based on these comparisons, we can consider that the Beard (1974) model is validated for addressing the collection of the aerosol particles of the accumulation mode by raindrops. Finally, it seems necessary to provide, to facilitate its use, an analytical expression to assess the contribution of the rear capture ($E_{Rear-capture}$) to the raindrop collection efficiency. Indeed, the Slinn (1977) model which neglects rear capture underestimates the collection efficiency by two orders of magnitude in the submicronic range compared to Beard's model (1974).

Furthermore, Beard (1974) noticed from his theoretical simulations that rear capture plays a main role in collection efficiency for aerosol particles with a Stokes number smaller than $5\times10^{-2}$. Thus, to derive an analytical expression for the elementary collection efficiency resulting from rear capture alone ($E_{rear\ capture}$), we gather in Figure 13 the collection efficiencies numerically simulated by Beard (1974) for a Stokes number smaller than

$5\times10^{-2}$ (crosses in Figure 13). These collection efficiencies are plotted as a function of the Reynolds number of the drops and the Stokes number of the particles.

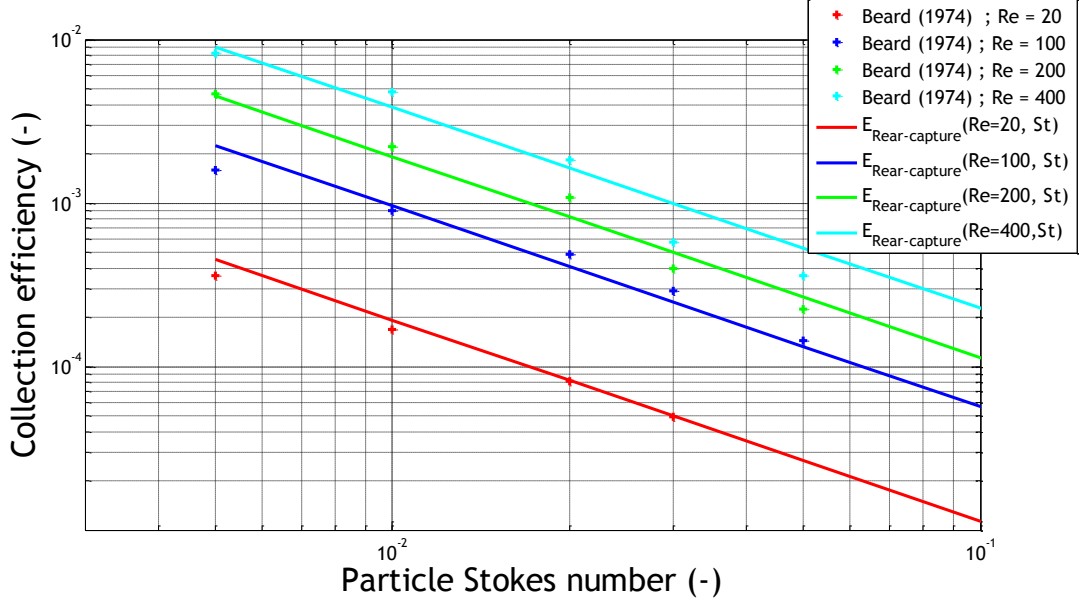

Figure 13. Semi-empirical parametrization of rear capture.

This figure suggests that the Reynolds number of the drop and Stokes number of the aerosol particles are the two parameters influencing rear capture. The dependency on these two dimensionless numbers is physical as the Reynolds number of the drop $(Re_{drop})$ reflects the intensity and the size of the areas of recirculating flow in its wake and the particle Stokes number $(St_{ap})$ reflects the susceptibility of the particle to pass through the recirculating flow in the wake of the drop without being trapped.

Applying a power law fit to the simulations of Beard (1974) yields equation 14.

$$E_{Rear-capture} = \frac{1}{3\times10^{7}}\ Re_{drop} \times St_{ap}^{-1.23} \quad (14)$$

This correlation is presented in solid lines in Figure 13 and shows a satisfactory agreement with K.V. Beard's simulations (crosses) in the corresponding range of drop Reynolds number and particle Stokes number. However, it should be kept in mind that this relationship is only valid for drop Reynolds numbers larger than 20

(a 280 μm drop at its terminal velocity), since below this critical value there is no recirculating flow behind the drop (Le Clair *et al.,* 1972). Finally, this new contribution should be added to those presented in Table 1 for raindrops.

**Conclusion**

This study is a follow up of the paper by Quérel *et al.* (2014b) and treats questions raised therein. In particular, Quérel *et al.* (2014b) showed that their efficiency measurements of submicron particles could only be explained
by rear capture. The present paper confirms the impact of recirculating flows at the rear of the drop on the collection of submicron particles. This was done by directly comparing our measurements against the numerical simulations of Beard (1974). The BERGAME experimental facility was optimised to considerably reduce the measurement uncertainties, as well as to perfectly control the electric charges of both the drops and the aerosol particles.

As in Quérel *et al.* (2014b), we show that the collection efficiency of the accumulation mode aerosol particles by drops representative of rain varies significantly with the size of the particles. On a logarithmic scale, the efficiency curve obtained shows a "V" shape with a minimum around 0.65 μm. The increase in collection efficiency for particles larger than 0.65 μm is attributed to the mechanism of impaction on the front side of the drop. Within this size range, the increase in the diameter of the particle increases its inertia, and the particle can
no longer follow the streamlines, and thus impacts the drop. It was not possible for the measurements of Quérel *et al.* (2014b), but here we can directly compare our results with the numerical simulations carried out by Beard (1974). This comparison highlights the robustness of his model for predicting the efficiency of capture of particles by raindrops over the entire accumulation mode. It should be noted that it is the only model to predict the significant increase in collection efficiency that we measured for submicron particles. This is related to the
fact that Beard (1974) first simulated the flow around the drop by solving the complete Navier-Stokes equation (without ignoring the convection terms; Beard and Grover, 1974). He, therefore, captured the separation of the boundary layer at the rear of the drop and the resulting recirculating flows; and then, he simulated the trajectory of the particles in this velocity field. K. V. Beard thus showed that the increase in the collection efficiency of submicron particles as observed in experiments is due to the fact that these particles are captured in the
recirculating flows to the rear of the drop and drawn back into its rear side.

Furthermore, we have also shown that, for particles larger than one micrometre, the models of K.V. Beard and W.G.N. Slinn are very similar. Finally, we propose a new semi-analytical expression to calculate the elementary efficiency of capture by the rear recirculating flows. It is important that this mechanism should be systematically taken into account to avoid errors of at least two orders of magnitude on the collection efficiency and
consequently on the scavenging coefficient.

In the near future, we plan to integrate these new measurements within the DESCAM model (Flossmann, 1986; Flossmann, 1991; Querel *et al.*, 2014a) and to compare the scavenging coefficient derived from the theoretical approaches and the experiments conducted in the environment by Volken and Shuman (1993), Laakso *et al.* (2003) and Chate (2005).

Finally, we plan, in a more distant future, to look at other hydrometeors such as snow and hail.

**Appendix 1: Evaluation of uncertainties**

5  The collection efficiency is calculated by means of equations (5) and (8) from which we derive the equation below, by substitution:

$$E(d_{aero}, D_{drop}, RH) = \frac{2\, D_{Drop}.[fluo]_{drop}.V_{chamber}}{3H.[fluo]_{filter} \cdot V_{sol}}$$

The expanded relative measurement uncertainty of the collection efficiency ($U_{R,E(d_{aero},D_{Drop},RH)}$) is determined with the help of the law of propagation of variances, considering an expansion factor of two (Lira, 2002) :

$$U_{R,E(d_{aero},D_{Drop},RH)} = 2\sqrt{u_{R,\,D_{drop}}^2 + u_{R,[fluo]_{drop}}^2 + u_{R,V_{chamber}}^2 + u_{R,H}^2 + u_{R,[fluo]_{filter}}^2 + u_{R,V_{sol}}^2}$$

In the right-hand member of this expression, the terms $u_{R,X}$ correspond to the relative measurement uncertainty of $X$. Each experimental uncertainty is discussed in a separate sub-section.

*Uncertainty in drop size*

Shadowgraph measurements of the size of the drops have shown that our drop generation system is very stable and reproducible for the parameters adopted (section 2.1). The standard deviation of the drop size distribution is 20 µm; we use this standard deviation to determine the relative uncertainty in the diameter of the drops.

$$u_{R,\,D_{drop}} = \frac{\sigma_{D_{drop}}}{D_{drop}} = \frac{20 \times 10^{-3}}{1.3} \approx 0.015$$

*Uncertainty in fluorescein concentration measurements*

For the range of concentrations within which fluorescence spectrometry is used, the calibration certificate of the
spectrometer indicates an expanded relative measurement uncertainty ($U_{R,[fluo]}$) of less than five percent.

We then derive the relative measurement uncertainty of the fluorescein concentration in the drops ($u_{R,[fluo]_{drop}}$). This relative uncertainty has two contributions. The first one is due to the spectrometer relative measurement uncertainty on the fluorescein concentration ($u_{R,[fluo]}$), and the second one is due to a potential variation of the volume of water collected, (in the drop collector, Figure 7), due to vaporization during the experiments
($u_{R,V_{collected}}$).

$$u_{R,[fluo]_{drop}} = \sqrt{\left(u_{R,[fluo]}\right)^2 + \left(u_{R,V_{collected}}\right)^2}$$

$$u_{R,[fluo]} = \frac{U_{R,[fluo]}}{2} = \frac{0.05}{2}$$

The uncertainty on the volume of water collected ($u_{R,V_{collected}}$) is estimated with the maximum variation of the volume of liquid water in the drop collector, due to vaporisation ($EMT_{V_{collected}}$).

$$u_{R,V_{collected}} = \frac{EMT_{V_{collected}}}{3\,V_{collected}}$$

In this equation, the volume of water collected ($V_{collected}$) is greater than one cubic centimetre (section 2.4). The maximum variation of the volume of liquid water in the drop collector ($EMT_{V_{collected}}$) is evaluated supposing that during the experiment period (section 2.4) the entire volume of the buffer ($V_{buffer}$) becomes saturated with water vapour. This leads to:

$$EMT_{V_{collected}} = \frac{3\,M_{H_2O}P_{sat}(T_{air})V_{buffer}}{RT_{air}\rho_{liquid-water}} = 1.2 \times 10^{-2} cm^3.$$

In this equation, R is the perfect gas constant, $P_{sat}$ is the saturation vapour pressure, $\rho_{liquid-water}$ is the density of liquid water, $M_{H_2O}$ is the molar mass of water and $T_{air}$ the gas temperature in the buffer. The three coefficient on the numerator comes from the fact that the buffer volume is flushed three times during the measurement period (section 2.4).

$$u_{R,[fluo]_{drop}} = \sqrt{\left(\frac{0.05}{2}\right)^2 + \left(\frac{1.2 \times 10^{-2}}{3 \times 1}\right)^2} \approx 0.025$$

For the fluorescein concentration measured in the aerosol chamber($[fluo]_{chamber}$), we have the same uncertainty associated with the fluorescence spectrometry measurement. In addition to this measurement uncertainty, there is a second uncertainty associated with the reduction in concentration during the course of the experiment. We have calculated this reduction to be less than eight percent over the duration of the measurement. The total relative uncertainty in the fluorescein concentration inside the aerosol chamber is therefore

approximately 8% (equation below).

$$u_{R,[fluo]_{chamber}} = \sqrt{0.025^2 + 0.08^2} \approx 0.08$$

### Uncertainty in height of aerosol chamber

The aerosol chamber measures 1.3 metres plus or minus 1 millimetre. However, over the duration of the measurement, the particles diffuse and move slightly outside the geometric boundaries of the aerosol chamber. We calculate the maximum error in the height of interaction between the drops and the particles ($EMT_H$) to be

approximately two centimetres (one above and one below the chamber). We therefore calculate the relative uncertainty for this height of interaction ($u_{R,H}$) by means of the following equation:

$$u_{R,H} = \frac{EMT_H}{3\,H} \approx 0.005$$

### Uncertainty in volume of dilution:

The uncertainty in the volume of dissolution is very low, we estimate its maximum error ($EMT_{V_{sol}}$) to be one

millilitre. We derive a relative uncertainty in the dilution ($u_{R,V_{sol}}$):

$$u_{R,V_{sol}} = \frac{EMT_{V_{sol}}}{3\,V_{sol}} \approx 0.003$$

## *Uncertainty in volume of aerosol chamber*

The uncertainty in the volume of the aerosol chamber is low, we estimate its maximum error ($EMT_{V_{chambre}}$) to be 20 centilitres. We derive the relative uncertainty in the dilution ($u_{R,V_{chamber}}$):

$$u_{R,V_{chamber}} = \frac{EMT_{V_{chamber}}}{3\,V_{chamber}} = \frac{20 \times 10^{-2}}{3 \times 10} \approx 0.007$$

## *Uncertainty in relative humidity*

The relative humidity is not directly involved in the calculation of collection efficiency. However, it is established, for the finest droplets, that the efficiency increases considerably when the relative humidity reduces, due to diffusiophoresis. For example, Grover *et al.* (1977) calculated that the collection efficiency of a 0.5 μm

aerosol particle by a 80 μm, can increase by a factor of $10^4$ when the relative humidity falls from 100 to 20%.

However, our recent measurements, for the largest hydrometeors forming rain (between 2 and 2.6 mm; Quérel *et al.,* 2014b) showed no dependency of the collection efficiency on relative humidity.

During our experiments, the aerosol generator settings were optimised in such a way that, at the end of the aerosol chamber filling phase, the relative humidity in the chamber was 75 ± 1%.

For each measurement, during the 10 minutes needed to collect one millilitre of drops (section 2), the relative humidity increased by 5 ± 1%. This increase is related to an accumulation of water on the slightly inclined bottom of the aerosol chamber.

We consider therefore that the measurement uncertainty for the relative humidity is approximately 5%.

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
