# Peer review of "Experimental evidence of the rear capture of aerosol particles by raindrops"

_Atmospheric Chemistry and Physics, 2016_

## Referee Comment (RC1) · Anonymous Referee #2 · 31 Jan 2017

**Major comments**

Droplets generated by means of piezo-elements might be charged, giving rise to electro-scavenging forces which can affect the collection rate: see, for example, Pranesha et al. (1966) and Ardon-Dryer et al. (2015). In addition, the aerosol neutralizer leads to a Boltzmann distribution where the most common charge state other than neutral is a single charge. Therefore electro-scavenging forces, due to spurious charges, cannot be ruled out unless drop charges are measured. In addition, the droplets are falling in a subsaturated environment (77% RH) and could therefore evaporates giving rise to phoretic forces.

The authors should clarify these important aspects in order to highlight their results.

Line 73. " *The Slinn model (1977) does not reproduce this increase efficiency, leading to errors of several order of magnitude*...". Lai et al (1978) measured scavenging of aerosol particles by falling water droplets. They compared their results against Beard and Slinn models. Concerning the Beard model the authors stated that "... *his estimation of the collection efficiency is more than one order of magnitude lower than our experimental results*..." and concerning Slinn model *"...hardly predicts the qualitative features of our results, it intercepts the range of values obtained in this work with submillimeter droplets ...*". Therefore Lai et al. (1978) concluded that the Slinn model at least intercepts their experimental data. The authors should explain this different conclusion.

Line 164 *"...the Slinn model underestimates by two order of magnitude the measured collection efficiency for submicron-sized particles*". The figure below shows, among others, the collection efficiency from Lai et al (1978), present manuscript, Beard (1974) and Slinn (1971), Quérel et al., (2014). The collection efficiencies are given as particle aerodynamic diameter. It can be seen that the experimental points from the present manuscript are in agreement with Beard (for D < 1 micron); however the collection efficiency from from Slinn model (taken from Fig. 6 of Lai et al. paper) is in better agreement with Lai and not with the manuscript authors (see previous comment).

[Figure]

Figure 1

The experimental points from Lai et al., were obtained with AgCl with a particle density of 5.6 g cm$^{-3}$, larger than the particle density used in the manuscript (1.3 g cm$^{-3}$).

Droplet rear particle capture is mainly due to inertial effects (Brownian diffusion is too weak in this particle size range) therefore, in order to compare results from different papers, the collection efficiency should be compared against aerodynamic particle diameter or even better against the Stokes number.

Line 478. "*The reasons for the increase in collection efficiency for particles smaller than 0.65 µm in diameter are not as easy*". This sentence is badly written since all scavenging models predict an increase in collection efficiency for submicronic aerosol particles due to Brownian and turbulent diffusion processes (among others Davenport and Peters 1978, Park et al., 2005). The authors should clarify that they are considering a limited size range where Brownian diffusion is not important.

Table 1. Since the input aerosol is not monodisperse, it is not clear how the authors report the collection efficiency for the particle sizes given in the table.

Table 1. $U_{R,E}$ is the relative measurement uncertainty which is mainly due to the contribution of fluorescein uncertainty inside the aerosol chamber (0.08). The propagation of variances equation (line 612) gives about 0.17. Table 1 (first raw) gives $4.5 \ 10^{-4}$ as the $U_{R,E}$ value. If it is the absolute uncertainty, then E times $U_{R,E}$ ($8.8 \ 10^{-3}$ times 0.17) gives $1.5 \ 10^{-3}$ not $4.5 \ 10^{-4}$ as reported. The authors should explain better the data shown in Table 1.

**Minor comments**

Line 30. *Beard (1974).* In the reference list there are two papers from Beard (1974).
Line 107. Mircea et al. is not in the references list.
The reference list is not typographically uniform.
Line 424. $M_{drop}$ becomes $M_{gtte}$ in equation 6 and so on.
The English language of the manuscript should be revised.

**References**
Ardon-Dryer K., et al., ACP, 15, 9159-9171, 2015.
Davenport H.M., et al., Atmos. Env., 12, 997-1008, 1978.
Lay K.Y., et al., J. Atmos.Sci., 35, 674-682, 1978.
Park S., et al., J. Aerosol Sci., 36, 1444-1458, 2005.
Pranesha T.S., et al., JGR, 101,D18, 23,373-23,380,1996.

---

## Referee Comment (RC2) · Anonymous Referee #1 · 1 Feb 2017

Scavenging of aerosols by raindrops has been a very important topic of atmospheric physics since the '80s. Recently it became actual again due to nuclear disasters. The major uncertainty of the efficiency of scavenging originates from the unknown collection/collision efficiencies between water drops and aerosol particles. Especially in the size range of raindrops the determination of this efficiency is difficult due to the wake capture caused by eddies behind the falling drops. The manuscript of Lemaitre et al. presents a possible experimental evidence for the wake capture, and provides parameterization to account for this effect. Nevertheless, this effect can also play a very important role in other atmospheric processes in which drop-particle collisions involved, such as contact freezing, for instance. In general the paper is clearly written and well organized, although I found the discussion of the novel parameterization and its relation to the experiments incomplete. The quality of the tables and figures is good. I really

appreciate the error discussion in the Appendix. The paper can be recommended for publication in Atmospheric Chemistry and Physics. Nevertheless, I list some minor comments that can be taken into account for a revision before publication:

p.1. line 31: "Aerosol particles are important components…" – please correct.

p. 1. Line 33: "our"should be read instead of "out"

p. 2. Line 34: You compare the results of your measurements against the model outcomes, but not the measurement itself. Please revise.

p. 6., Fig. 2: Pressurized should be written instead of Pressurisez

p. 7. Line 6: What does S_goutte stay for?

p. 7. Eq. 4: Is this formula equivalent with the integration method of Müller et al. ? If no, how is it related to that? What is the accuracy of the size determination using Eq. 4?

p. 7. Figure 4: The caption is actually wrong. The figure does not show the setting parameters, such as pressure, opening time, but the generated drop diameters using a predefined pressure and different opening times. Please revise the caption.

p. 8. Line 2: The sentence should be reformulated.

p. 8. Line 15: I believe the fall distance to reach terminal velocity was sufficient in the experiments, which could/should also been shown here by comparing the result with the theoretical calculations of Wang and Pruppacher (1977).

p. 8. Line 17: How did you determine the axis ratio? Is it just the ratio of the vertical and horizontal dimensions of the drop? Do you see any canting of the drops, or are they perfectly horizontally aligned?

p. 9: Line 18: APS; ELPI: online methods; filter: offline method. Can you compare their results?

p. 10. Figure 7: I think some labels and arrows are shifted in the Figure.

p. 10. Line 9: How does argon help to minimize sedimentation? I think it would be useful to explain it in somewhat more detail.

p. 11. Line 15: The sentence should be revised.

p. 11. Line 24: What are the detection limits of the methods? I suggest to indicate these detection limits also in the figure?

p. 12. Line 29: Within the 10 minutes measurement time the collected drops might evaporate. How do you deal with this water loss in the collector?

p. 13. Figure 9: Where is the buffer volume? I suppose it is between the drop collector and the knife gate valve, right? It should be shown in the figure in this way.

p. 13. Line 20: You claim that you measured the stability of the aerosol generator. What was the result of it? It is not indicated.

p. 13. Eq 6. What do here $M\_gtte$, and $[fluo]\_drop$ stay for?

p. 14. Eq. 7: What is $H$ in the equation?

p. 14. Line 3: Eq. 8 is just the ratio between Eq. 6 and 7; this should be noted again.

p. 14. Line 8-9: The word "same" is unnecessarily often used.

p. 14. Line 13: I believe you mean here the resolution of the instruments. It would therefore be worthwhile to give some specifications of them; such as detection limit (see one of my earlier comments), resolution, etc.

p. 14. Table 1: The uncertainties of the measured quantities should be indicated.

p. 14. Eq. 9: How do you determine the Cunningham factor for $d\_ap$? Is this an iteration method to calculate $d\_ap$?

p. 14. Eq. 10: What is FG in the equation?

[Figure]

p. 15. Line 15: "x axis" is a very loose formulation.

p. 15. Line 18: "model of reference in the environment" – again a loose formulation.

p. 15. Line 24: I suggest here to consider the other physical processes involved in collision for different particle sizes, such as Brownian motion, thermophoresis, diffusiophoresis, electroscavenging, etc. Good reference for that might be the paper of Ladino et al., ACP, pp. 9745 (2013), and the book of Pruppacher and Klett (2010).

p. 17. Line 10: To be honest, I do not see the significant improvement. The difference between measurements and theory is still large. I suggest you to indicate a range of possible collection efficiencies as a shadowed area, for instance, in the Figure by calculating collection efficiencies corresponding to the smallest and largest aerosol particle at a given d_ap.

p. 17. Line 18: I do not really get the point. Have you taken into account the rear capture or not? If not, is it possible to do that and modify the collection efficiency curve?

p. 18. Line 4: I do not see the relevance of this figure, and its connection to the present experiments. Again, how do you account for the rear capture to calculate the collection efficiency?

p. 20. Line 7: Please use dot instead of comma for the numbers.

p. 21. Line 3: The dimension of the volume (and, consequently, its error) is meter cube, nor meter.

---

## Author Comment (AC1) · 4 Mar 2017

Dear reviewer,
Please find the point by point answer to your questions
To facilitate the discussion I adopted a colour code:

- In red are the questions
- Blue are the answers
- Green are the modifications or additions in the article

Thank you for taking your time for the review
Kind regards

**Major comments**

Droplets generated by means of piezo-elements might be charged, giving rise to electro-scavenging forces which can affect the collection rate: see, for example, Pranesha et al. (1966) and Ardon- Dryer et al. (2015). In addition, the aerosol neutralizer leads to a Boltzmann distribution where the most common charge state other than neutral is a single charge. Therefore electro-scavenging forces, due to spurious charges, cannot be ruled out unless drop charges are measured. In addition, the droplets are falling in a subsaturated environment (77% RH) and could therefore evaporates giving rise to phoretic forces.

The authors should clarify these important aspects in order to highlight their results.

In order to better introduce the various mechanisms involved in the collection of the aerosol particles by drop (and droplets) we added in the first section after the introduction (Theoretical description of washout) a paragraph describing all the mechanisms involved in the collection of the aerosol particles by raindrops. (p4, line 1-30).

"Several mechanisms are usually considered to simulate the collision between aerosol particles and droplets. We recall them briefly, however, a more exhaustive review can be found in the literature (Pruppacher et al., 1998; Chate, 2005; Ladino et al., 2013; Ardon-Dryer et al., 2015). The three main mechanisms leading to this collection are Brownian motion, inertial impaction and interception. Small particles with a radius on the order of the mean free path or smaller are very sensitive to the collision of air molecules and scatter from streamlines of the flow due to Brownian motion. For large particles with a diameter greater than 1 μm, their inertia prevents them from following the streamlines of the flow and they impact the drop on its leading edge. Aerosol particles with a diameter smaller than 1 μm and much larger than the mean free path of the air molecules follow the streamlines of the flow around the drop. They might nevertheless enter in contact with the drop because the streamlines are approaching the drop at a distance smaller than the radius of the aerosol particle. For particles with diameter between 0.2 μm and 1 μm, there is a minimum collection efficiency called the "Greenfield Gap" (Greenfield, 1957). For these particles phoretic forces are expected to be the most efficient mechanisms. Thermophoresis and diffusiophoresis are respectively linked to thermal and water vapour gradients. The side of the aerosol particles exposed to the warmer air is impacted by molecules whose kinetic energy is higher than that of the molecules impacting the colder side of the particles. As a result, thermophoresis results in a force whose direction is the opposite of the thermal gradient. Similarly, particles exposed to a water vapour gradient are exposed to molecular collisions with a dissymmetric kinetic energy since water vapour molecules are lighter than

air molecules. In the atmosphere, diffusiophoresis thus results in a force whose direction is the opposite of the water vapour gradient. Electro-scavenging could also have an important contribution when both droplets and aerosols particles are electrically charged, resulting in an attractive (or repulsive) force when they have opposite (or the same) polarity. Moreover, Tinsley et al. (2000, 2006) theoretically showed that electrically charged aerosol particles can induce an image charge on droplets that results in a short range electrical attraction that increases collection efficiency even with neutrally charged droplets.

For each of these elementary mechanisms, theoretical expressions of the elementary collection efficiencies have been derived (Table 1).

Table 1. References of theoretical expressions for the calculation of each collection mechanism

| Elementary mechanism | Reference |
|---|---|
| Inertial impaction | Slinn (1977); Park et al. (2005) |
| Interception | Slinn (1977); Park et al. (2005) |
| Brownian motion | Slinn (1977); Park et al. (2005) |
| Diffusiophoresis | Waldmann (1959); Davenport and Peters (1978); Andronache et al. (2006);  Wang et al. (2010) |
| Thermophoresis | Davenport and Peters (1978); Andronache et al. (2006); Wang et al. (2010) |
| Electro-scavenging | Davenport and Peters (1978); Andronache et al. (2006); Wang et al. (2010) |
| Image forces | Tinsley and Zhou (2015) |

Finally, the droplet total collection efficiency can be theoretically deduced by adding all these elementary collection efficiencies together. The use of these theoretical models seems justified for cloud droplets since they have very small Reynolds numbers. However, there are many uncertainties concerning raindrop size range."

Once all the mechanisms described, we explain that we want to compare to Beard (1974) model, and thus to minimize all the mechanisms he did not considered in his simulations. (p7 line 2-6).

"The objective of these modifications is also to be consistent with the hypothesis of the Beard (1974) model, which considers only drag and gravitational forces on the aerosol particles. The modifications are thus intended to minimise electro-scavenging (discussed in sections 2.1 and 2.3), diffusiophoresis (discussed in section 2.3 and Appendix 1) and thermophoresis. Both the drop generator and aerosol chamber are described in the following sections. "

-Concerning electro-scavenging:

We minimized electo-scavenging by checking that the drop are not electrically charged. The drop charges are measured with the help of a Faraday pail connected to an electrometer (Keithley model 6514; Sow & Lemaitre, 2016). Any electrical charge on the drop could not be measured even with the high sensitivity of the electrometer (10 fC) and even integrating the measured charge on a large number of drops. It should be noted that this generator is completely different than microdrop Technologies and MicroFab Technologies), first the piezo element is not in contact with the fluid. Second the all hydraulic system is grounded.

We add the following paragraph in section 2.1 (P7 line 23 to p8 line 2).

"Classical piezoelectric drop-on-demand systems may produce electrically charged droplets (Ardon-Dryer et al., 2015). However, we want to limit electro-scavenging as Beard (1974) did in his simulations.

Thus, the net charge of each drop produced by this system has been measured with the help of a Faraday pail connected to an electrometer (Keithley model 6514; Sow & Lemaitre, 2016). Any electrical charge on the drop could not be measured even with the high sensitivity of the electrometer (10 fC). This might be explained by the fact that unlike classical piezoelectric drop-on-demand systems (such as those of microdrop Technologies and MicroFab Technologies), the piezoelectric transducer in our drop generator is not in direct contact with the liquid (Figure 2)."

Finally the aerosols particles are neutralized. Thus they have a Boltzmann charge distribution.

-Concerning diffusiophoresis:

To be sure that during our experiment (with 77 % relative humidity) diffusiophoresis is a second order mechanism, we compared the elementary collection efficiency it induces compared to beard (1974) model. Diffusiophoresis was calculated with the model of Davenport & Paters (1978). This comparison is presented in figure 12.

We added the following tex in section 2.3( p13, line 34):

Furthermore, at this high relative humidity, diffusiophoresis is not expected to contribute significantly to the collection efficiency, even close to the minimum of efficiency. Indeed, the contribution of diffusiophoresis calculated with the model of Davenport and Peters (1978) for our experimental conditions (relative humidity, air temperature and drop size) is $2.5 \times 10^{-4}$, which is smaller than the collection efficiencies predicted by the Beard model (Figure 1).

Line 73. " *The Slinn model (1977) does not reproduce this increase efficiency, leading to errors of several order of magnitude*...". Lai et al (1978) measured scavenging of aerosol particles by falling water droplets. They compared their results against Beard and Slinn models. Concerning the Beard model the authors stated that "... *his estimation of the collection efficiency is more than one order of magnitude lower than our experimental results*..." and concerning Slinn model *"...hardly predicts the qualitative features of our results, it intercepts the range of values obtained in this work with submillimeter droplets ...*". Therefore Lai et al. (1978) concluded that the Slinn model at least intercepts their experimental data. The authors should explain this different conclusion.

This point is discussed together with next question. Nevertheless on point is still strange for us, because when we compare Beard (1974) and Slinn (1977) model, we more or less, find the same results for aerosol particles with stokes number  greater than 0.05, however Lai find some differences.  May be Lai (1978) added diffusiophoresis to their calculations as it seems that their experiments are dominated by diffusiophoresis. However we don't have many details on that point on the publication from Lai *et al*.  (1977).

Line 164 *"...the Slinn model underestimates by two order of magnitude the measured collection efficiency for submicron-sized particles"*. The figure below shows, among others, the collection efficiency from Lai et al (1978), present manuscript, Beard (1974) and Slinn (1971), Quérel et al., (2014). The collection efficiencies are given as particle aerodynamic diameter. It can be seen that the experimental points from the present manuscript are in agreement with Beard (for D < 1 micron); however the collection efficiency from from Slinn model (taken from Fig. 6 of Lai et al. paper) is in better agreement with Lai and not with the manuscript authors (see previous comment). Figure 1The experimental points from Lai et al., were obtained with AgCl with a particle density of 5.6 gcm-3, larger than the particle density used in the manuscript (1.3 g cm-3). Droplet rear particle capture is mainly due to inertial effects (Brownian diffusion is too weak in this particle size range) therefore, in order to compare results from different papers, the collection efficiency should be compared against aerodynamic particle diameter or even better against the Stokes number.

We explored different hypothesis to explain the difference between Lai (1978) measurements and ours.
- First it seems that the particle they use have same geometric standard deviation as ours, thus the difference could not come from this point ("*For a particular aerosol the particular size distribution typically had a variance of about 10 % from the modal value*").
- Second, it seems that the particles they use are not neutralised by any system. However it seems that they are more or less neutralised. Indeed when they plot the influence of the drop charge on the collection efficiency (Figure 7 from their article) their measurements are symmetric with the ordinate axis.
- It seems that their drop are not exactly at terminal velocity. Indeed they used a 455 cm shaft to accelerate their drop however it seems from Wang, & Pruppacher, (1977b) that 6.5 m are needed to reach 99% of the terminal velocity. (we have 8m). but we think this is not the point that explain of the difference between Lai results and ours.
- The most convincing reason explaining this difference is the relative importance of diffusiophoresis. Indeed Lai performed their experiments in pure nitrogen. Thus their experiments seem to be driven by diffusiophoresis.

To compare our measurements with Lai's ones we need to plot them as function of the Stokes number as advised by the reviewer. Then we compared Beard (1974) and Slinn (1977) models together with present and Lai's results (figure 12). The analysis of Lai's experimental procedure shows that their aerosol chamber is filled with pure Nitrogen. Thus we had on the figure the contribution of diffusiophoresis calculated from the model of Davenport & peters (1978) in our and Lai experiment. This highlights that Lai experiments are dominated by diffusuiphoresis. This is the reason why they doesn't fit beard results.

We added : (p 20 l 15 - p 21 l 23)

"Furthermore, it is interesting to compare our measurements with the ones from Lai *et al.* (1978) since they are the only ones in the literature in the same drop size range. As the aerosol particles produced in these experiments are composed of silver chloride ($\rho_{AgCl} = 5.6 \ g.cm^{-3}$), which is much denser than sodium fluorescein ($\rho_{C_{10}H_{10}Na_2O_5} = 1.3 \ g.cm^{-3}$), it is more appropriate to plot all the collection efficiencies as a function of the Stokes number of the particle ($St_{ap}$).

$$St_{ap} = \frac{\rho_p U_\infty (D_{drop}) d_{ap}^2 C_{c,d_{ap}}}{9 D_{drop} \mu_{air}}$$

In this equation, $\mu_{air}$ is the dynamic viscosity of the air and $\rho_p$ the density of the aerosol particles. This comparison is presented on Figure 12.

[Figure]

Figure 1. Comparison of our measurements with Lai *et al.* (1978). Both measurements are compared with the Slinn (1977) and Beard (1974) models. The contributions of diffusiophoresis are evaluated in both experiments with the model of Davenport and Davis (1978)

For particles with a Stokes number greater than $6\times10^{-2}$, the motion of the particles is driven by their inertia, leading us to expect to observe the same trends in our measurement and those of Lai *et al.* (1978). The comparison for Stokes number smaller than $6\times10^{-2}$ is much less obvious. Indeed, for these particles, the measurements of Lai *et al.* (1978) indicate an increase in collection efficiency, while our measurements continue to decrease down to a Stokes number of $1.6\times10^{-2}$. At that point, the slopes of the increases of both collection efficiency measurements are similar, while the Stokes number decreases.

A precise analysis of the procedure for the aerosol particle injection in the experiments of Lai *et al.* (1978) indicates that the carrier gas is pure nitrogen without any subsequent humidification. As a consequence, it is reasonable to consider that their measurements were performed with 0% relative humidity. In order to compare the contribution of diffusiophoresis for both our experiment and that of Lai *et al.* (1978), we plot in Figure 12 the elementary contribution of diffusiophoresis ($E_{dph}$) to the collection efficiency. This contribution is calculated with the Peters and Davenport (1978) model for 0% relative humidity (as expected for the experiments of Lai *et al.*, 1978) and 77% (as measured in our experiments). From this figure it will be noted that for the experiments of Lai *et al.* (1978), the contribution of diffusiophoresis is more than one order of magnitude higher than in ours. Furthermore, while in our experiments the contribution of diffusiophoresis is smaller than the collection efficiency simulated by Beard (1974), the opposite is observed with Lai *et al.* (1978). Thus, it appears that the

experiments of Lai *et al.* (1978) cannot be compared directly to Beard (1974)'s model, because they seem to be dominated by diffusiophoresis."

Line 478. "*The reasons for the increase in collection efficiency for particles smaller than 0.65 μm in diameter are not as easy*". This sentence is badly written since all scavenging models predict an increase in collection efficiency for submicronic aerosol particles due to Brownian and turbulent diffusion processes (among others Davenport and Peters 1978, Park et al., 2005). The authors should clarify that they are considering a limited size range where Brownian diffusion is not important.

In order to better introduce the mechanisms leading to the collision of aerosol particles with drops we added a paragraph that introduces all the mechanism (p4, line 1-30).
Moreover we added a small sentence to say that Brownian diffusion is not expected to play an important contribution in the particle size range investigated: (p 19 line 17-18)

"The reasons for the increase in collection efficiency for particles smaller than 0.65 μm in diameter are not as easy to figure out. Indeed particles of this size range are not expected to be affected by Brownian motion since their diameter is seven times bigger than the mean free path of the air molecules. "

Table 1. Since the input aerosol is not monodisperse, it is not clear how the authors report the collection efficiency for the particle sizes given in the table.
We add just below the table:
"In this table, the aerosol diameter ($d_{aero}$) is the median aerodynamic diameter of each particle size distribution measured using the APS or the ELPI."

Table 1. UR,E is the relative measurement uncertainty which is mainly due to the contribution of fluorescein uncertainty inside the aerosol chamber (0.08). The propagation of variances equation (line 612) gives about 0.17. Table 1 (first raw) gives 4.5 10-4 as the UR,E value. If it is the absolute uncertainty, then E times UR,E (8.8 10-3 times 0.17) gives 1.5 10-3 not 4.5 10-4 as reported. The authors should explain better the data shown in Table 1.

This is an error from me. The table has been modified to give the uncertainty with all the measurement discussed in appendix.

**Minor comments**
Line 30. *Beard (1974).* In the reference list there are two papers from Beard (1974). Modified
Line 107. Mircea et al. is not in the references list. Modified
The reference list is not typographically uniform. Modidied

Line 424. Mdrop becomes Mgtte in equation 6 and so on. Modified

The English language of the manuscript should be revised. The article has been rephrased and an English native speaker has correct the entire manuscript

**References**

Ardon-Dryer K., et al., ACP, 15, 9159-9171, 2015.

Davenport H.M., et al., Atmos. Env., 12, 997-1008, 1978.

Lay K.Y., et al., J. Atmos.Sci., 35, 674-682, 1978.

Park S., et al., J. Aerosol Sci., 36, 1444-1458, 2005.

Pranesha T.S., et al., JGR, 101,D18, 23,373-23,380,1996

---

## Author Comment (AC2) · 4 Mar 2017

Dear reviewer,

Please find the point by point answer to your questions

To facilitate the discussion I adopted a colour code:

- In black are your questions
- In Blue are the answers
- In Green are the modifications or additions in the article

Thank you for taking your time for the review

Kind regards

Q1 : p.1. line 31: "Aerosol particles are important components. . ." – please correct.

This is modified

Q2 : p. 1. Line 33: "our"should be read instead of "out"

This is modified

Q3 : p. 2. Line 34: You compare the results of your measurements against the model outcomes, but not the measurement itself. Please revise.

This is modified

Q4 : p. 6., Fig. 2: Pressurized should be written instead of Pressurisez

This is modified

Q5 : p. 7. Line 6: What does S_goutte stay for?

This is modified

Q6 : p. 7. Eq. 4: Is this formula equivalent with the integration method of Müller et al. ? If no, how is it related to that? What is the accuracy of the size determination using Eq. 4?

I don't know what is the integration method of Müller et al.. However we added details on the experimental setup, its resolution, and the hypothesis behind this equation (axisymmetric drops).

We write (P 8 line 8 to 27)

"For each opening time, shadowgraph measurements were taken in the aerosol chamber of the BERGAME facility. An optical window is used to trigger the photographing of each drop entering the BERGAME aerosol

chamber. Our optical device is a camera (Andor: neo, sCMOS) with a resolution of 2560 × 2160 pixels². It is equipped with a Canon macro lens (MP-E 65mm f/2.8 1-5x) for a magnification of 3:1 (experimentally checked with a calibration chart). The pixel size is 6.5 µm, for a spatial resolution of 2.1 µm. Drops are backlighted with a 9 ns strobe to freeze their fall on the sensor. An example of a shadowgraph image is shown in Figure 3.

[Figure]

Figure 1. Example of a shadow image

Due to the oscillations, the millimetric drops exhibit an oblate spheroid shape. To define the size of the raindrops the notion of "diameter equivalent to a sphere of the same volume" has been adopted. Since shadowgraphy yields only a 2-D information, the diameters are equivalent to a disc. For axisymmetric objects, volume and surface equivalent diameter are equal. Szakáll *et al.* (2009) experimentally verified this axisymmetric of drop of that size range at terminal velocity. Thus, shadow images are used and processed to deduce the projected surface area of the drop ($S_{drop}$) and derive the diameter of the disc of equal surface area ($D_{eq}$).

$$D_{eq} = \sqrt{\frac{4\,S_{drop}}{\pi}} \qquad (4) \qquad «$$

Q7 : p. 7. Figure 4: The caption is actually wrong. The figure does not show the setting parameters, such as pressure, opening time, but the generated drop diameters using a predefined pressure and different opening times. Please revise the caption.

We changed the caption to :

"Figure 2. Measured equivalent diameter of the drop produced by our generator as a function of the valve opening time (for an over pressure of 0.3 bar) "

Q8 : p. 8. Line 2: The sentence should be reformulated.

Reformulated :

"In order to be representative of rain the drops must cross the BERGAME aerosol chamber at their terminal velocity"

Q9 : p. 8. Line 15: I believe the fall distance to reach terminal velocity was sufficient in the experiments, which could/should also been shown here by comparing the result with the theoretical calculations of Wang and Pruppacher (1977).

We added the comparison in the article (p10 line 1 to 10).

"We note in this figure that up to a drop diameter of 1.4 mm, the 8 m distance is sufficient to accelerate the drops to their terminal velocity. This is consistent with the results of the theoretical calculations of Wang and Pruppacher (1977b), which predict that 6.5 m free fall is enough for a 1.4 mm drop to reach 99% of terminal velocity."

Indeed the Wang model was used to dimension our free fall shaft.

Q10 : p. 8. Line 17: How did you determine the axis ratio? Is it just the ratio of the vertical and horizontal dimensions of the drop? Do you see any canting of the drops, or are they perfectly horizontally aligned?

No canting was observed in the drop size range investigated in the article. The axis ratio is calculated as the ratio between the vertical and horizontal dimensions of the drop. We wrote (p 10 line 8 to 10):

"For the drop sizes investigated, drop can be considered as horizontally aligned oblate spheroids (Figure 3), no tilt angle was measured, which is consistent with Pruppacher & Beard (1970) measurements. This is why, the axis ratio is computed as the ratio between the vertical and horizontal dimensions of the drop. "

Q11 : p. 9: Line 18: APS; ELPI: online methods; filter: offline method. Can you compare their results?

This comparison performed on each experiment and is quite good: less than 10 % difference. We decided to keep the filter measurement as a reference to calculate the collection efficiency because in minimise the uncertainties. Indeed, the use of APS an ELPI measurement would induce additional hypothesis on the purity of the fluorescein salt (the supplier guarantees a purity of almost 97%) and the perfect sphericity.of the aerosol particles.

"For each of the particle sizes produced, the fluorescein mass concentrations in the aerosol chamber derived from APS and ELPI measurements are compared with ones derived from filter measurements (section 2.2). These comparisons provide slight differences (~10%) that can be attributed to both the purity of fluorescein sodium salt used (~97%) and the shape of the aerosol particles that is not perfectly spherical. Thus, for

improving the accuracy of collection efficiency measurements, the fluorescein concentration inside the aerosol chamber is derived from filter measurements, and APS and ELPI are used to provide a precise measurement of the particle size."

**Q12 :** p. 10. Figure 7: I think some labels and arrows are shifted in the Figure.

This point has been corrected

**Q13 :** p. 10. Line 9: How does argon help to minimize sedimentation? I think it would be useful to explain it in somewhat more detail.

The first experiments were performed without this argon layer. They showed a fast settling of the particles in the drop collector. This settling was order of magnitudes faster than the settling velocity of individual aerosol particles. A literature review on this point indicated that this fast "cloud settling" could be induced by Rayleigh-Taylor instabilities (Hinds *et al.*, 2002). These instabilities arise when a dense layer overlies a lighter one. As Argon is very dense this phenomena is not observer if the aerosol cloud overlies argon.

Using the layer of argon in the buffer volume allowed keeping the drop collector clean of particles for a period compatible with the experiments.

We wrote (p12 line 9 to 11) :

"One of the principal difficulties of these experiments relates to the sedimentation of the cloud of particles that settles directly inside the drop collector. Indeed, Rayleigh-Taylor instabilities can arises when a dense cloud of aerosol particles overlies a layer of clean air. These instabilities induce a downward motion of the aerosol cloud much faster that the settling velocity of individual particles (Hinds et al., 2002). In order to avoid this effect, a layer of argon (which is denser than the cloud of particles) is formed in the bottom of the aerosol chamber, located below the second gate valve in **Erreur ! Source du renvoi introuvable.**7."

**Q14 :** p. 11. Line 15: The sentence should be revised.

This typo is corrected:

"Changing the concentration of the solute dissolved in the water varies the size of the produced particles

**Q15 :** p. 11. Line 24: What are the detection limits of the methods? I suggest to indicate these detection limits also in the figure?

We introduced the measurement method of both APS and ELPI spectrometers at the beginning of section 2.3 (p13, line 4 to 19). Moreover we added on figure 7 the measurement range and the resolution of both instruments on figure 7.

"The aerosol particles size distributions are measured using an Electrical Low Pressure Impactor (ELPI, δ) and an Aerodynamic Particle Sizer (APS, χ).

ELPI is a quasi-real-time aerosol spectrometer (Marjamäki et al., 2000). It is composed of a corona charger and a 12-stage cascade low pressure impactor. Each stage of the impactor is connected to an electrometer. The corona charger is used to set the electrical charge of the particles to a specific level. Then, the low pressure impactor classifies the aerosol particles into 12 size classes according to their aerodynamic diameter (from 7 nm to 10 µm). Finally, the electrometers measure the electrical charge carried by the particles collected by each impaction stage. This charge is finally converted to the number of particles collected according to the charging efficiency function of the corona charger.

APS is also a quasi-real-time aerosol spectrometer (Baron, 1986). It measures the time-of-flight of individual particles accelerated by a controlled accelerating flow imposed by a calibrated nozzle. The time-of-flight of each aerosol particle is then converted into its aerodynamic diameter. Thus, the APS classifies the aerosol particles in terms of aerodynamic diameter from 500 nm to 20 µm over 52 size classes.

APS and ELPI are both used for their complementary size ranges so all the particles produced in our laboratory can be sized. For particles with a median aerodynamic diameter less than 0.8 µm, the size distribution is measured using an ELPI. For the others, we favour the use of an APS because of the better size resolution. "

**Q16 :** p. 12. Line 29: Within the 10 minutes measurement time the collected drops might evaporate. How do you deal with this water loss in the collector?

This point is discussed in appendix 1. We added this potential water evaporation to the global uncertainty on the fluorescein concentration in the drop.

We then derive the relative measurement uncertainty of the fluorescein concentration in the drops ($u_{R,[fluo]_{drop}}$). This relative uncertainty has two contributions. The first one is due to the spectrometer relative measurement uncertainty on the fluorescein concentration ($u_{R,[fluo]}$), and the second one is due to a potential variation of the volume of water collected, (in the drop collector, Figure 7), due to vaporization during the experiments ($u_{R,V_{collected}}$).

$$u_{R,[fluo]_{drop}} = \sqrt{\left(u_{R,[fluo]}\right)^2 + \left(u_{R,V_{collected}}\right)^2}$$

$$u_{R,[fluo]} = \frac{U_{R,[fluo]}}{2} = \frac{0.05}{2}$$

The uncertainty on the volume of water collected ($u_{R,V_{collected}}$) is estimated with the maximum variation of the volume of liquid water in the drop collector, due to vaporisation ($EMT_{V_{collected}}$).

$$u_{R,V_{collected}} = \frac{EMT_{V_{collected}}}{3\,V_{collected}}$$

In this equation, the volume of water collected ($V_{collected}$) is greater than one cubic centimetre (section 2.4). The maximum variation of the volume of liquid water in the drop collector ($EMT_{V_{collected}}$) is evaluated supposing

that during the experiment period (section 2.4) the entire volume of the buffer ($V_{buffer}$) becomes saturated with water vapour. This leads to:

$$EMT_{V_{collected}} = \frac{3\, M_{H_2O}P_{sat}(T_{air})V_{buffer}}{RT_{air}\rho_{liquid-water}} = 1.2 \times 10^{-2} cm^3.$$

In this equation, R is the perfect gas constant, $P_{sat}$ is the saturation vapour pressure, $\rho_{liquid-water}$ is the density of liquid water, $M_{H_2O}$ is the molar mass of water and $T_{air}$ the gas temperature in the buffer. The three coefficient on the numerator comes from the fact that the buffer volume is flushed three times during the measurement period (section 2.4).

$$u_{R,[fluo]_{drop}} = \sqrt{\left(\frac{0.05}{2}\right)^2 + \left(\frac{1.2 \times 10^{-2}}{3 \times 1}\right)^2} \approx 0.025$$

Q17 : p. 13. Figure 9: Where is the buffer volume? I suppose it is between the drop collector and the knife gate valve, right? It should be shown in the figure in this way.

Figure 9 is modified to show what we call the buffer volume. Moreover we added a small text just before the figure:

"At the end of these 200 seconds phases, the gate valves are closed again and the buffer volume between gate valve φ and the drop collector is flushed with argon (**Erreur ! Source du renvoi introuvable.**)."

Q18 : p. 13. Line 20: You claim that you measured the stability of the aerosol generator. What was the result of it? It is not indicated.

This was an imprecise formulation we verified the reproducibility. It was checked by reproducing several times the same procedure for the aerosol injection and comparing the characteristics of the aerosol particles (size, number and especially the concentration), at the end of the relaxation phase. Once we verified that same injection procedures give rise to reproducible initial conditions (with differences smaller than the uncertainty on the fluorescence spectrometer), we measurements the mass concentration of the aerosol particles but 15 min after the end of the relaxation phase. This showed a decrease oh 8% of the concentration during the all experiment. This 8 % decrease is the main source of uncertainty of our measurements (Appendix 1).

"For this, we have first verified the reproducibility of characteristics of the aerosol produced by the aerosol generator, in size, number and concentration. This is performed by repeating the injection phase with exactly the same operating conditions. No variation of the fluorescein concentration greater than the uncertainty of the fluorimeter ( $\pm 2.5\%$ , appendix 1) has ever been measured"

Q19 : p. 13. Eq 6. What do here M_gtte, and [fluo]_drop stay for?

This is corrected:

"The mass of fluorescein in the drops during the experiments $(M_{drop})$ is easy to calculate:

$$M_{drop} = \frac{\pi D_{drop}^3}{6} [fluo]_{drop} \quad (6)$$

where $[fluo]_{chamber}$ is the mass concentration of fluorescein in the aerosol chamber and $H$ the height of the aerosol chamber (1.3 m, Figure 1)."

Q20: p. 14. Eq. 7: What is H in the equation?

The definition of H I added in the test bellow equation 7 and also on figure 1.

"… and $H$ the height of the aerosol chamber (1.3 m, Figure 1)."

Q21 : p. 14. Line 3: Eq. 8 is just the ratio between Eq. 6 and 7; this should be noted again.

This remark is taken into account:

$$\text{"}E\big(d_{aero}, D_{drop}, RH\big) = \frac{M_{drop}}{M_2} = \frac{2\,D_{drop}.[fluo]_{drop}}{3H.[fluo]_{chamber}} \quad (8)\text{"}$$

Q22 : p. 14. Line 8-9: The word "same" is unnecessarily often used.

Corrected

Q23 : p. 14. Line 13: I believe you mean here the resolution of the instruments. It would therefore be worthwhile to give some specifications of them; such as detection limit (see one of my earlier comments), resolution, etc.

This remark is taken into account togather with question 15. We added a paragraph at the beginning of section 2.3.

"The aerosol particles size distributions are measured using an Electrical Low Pressure Impactor (ELPI, $\delta$) and an Aerodynamic Particle Sizer (APS, $\chi$).

ELPI is a quasi-real-time aerosol spectrometer (Marjamäki et al., 2000). It is composed of a corona charger and a 12-stage cascade low pressure impactor. Each stage of the impactor is connected to an electrometer. The corona charger is used to set the electrical charge of the particles to a specific level. Then, the low pressure impactor classifies the aerosol particles into 12 size classes according to their aerodynamic diameter (from 7 nm to 10 µm). Finally, the electrometers measure the electrical charge carried by the particles collected by each impaction stage. This charge is finally converted to the number of particles collected according to the charging efficiency function of the corona charger.

APS is also a quasi-real-time aerosol spectrometer (Baron, 1986). It measures the time-of-flight of individual particles accelerated by a controlled accelerating flow imposed by a calibrated nozzle. The time-of-flight of each aerosol particle is then converted into its aerodynamic diameter. Thus, the APS classifies the aerosol particles in terms of aerodynamic diameter from 500 nm to 20 μm over 52 size classes.

APS and ELPI are both used for their complementary size ranges so all the particles produced in our laboratory can be sized. For particles with a median aerodynamic diameter less than 0.8 μm, the size distribution is measured using an ELPI. For the others, we favour the use of an APS because of the better size resolution. "

**Q24 :** p. 14. Table 1: The uncertainties of the measured quantities should be indicated.

This has beed corrected

**Q25 :** p. 14. Eq. 9: How do you determine the Cunningham factor for d_ap? Is this an iteration method to calculate d_ap?

Yes it is. We added in the text

"This median aerodynamic diameter is converted into a physical diameter ($d_{ap}$) by means of the following expression (which is solved iteratively):

$$d_{ap} = d_{aero} \sqrt{\frac{C_{c,d_{aero}}}{C_{c,d_{ap}}} \left(\frac{\rho_0}{\rho_p}\right)} \qquad (9)$$ "

**Q26 :** p. 14. Eq. 10: What is FG in the equation?

This was a typo. It is corrected into GF

**Q27 :** p. 15. Line 15: "x axis" is a very loose formulation.

This is rephrased (p 16 line 9to 10)

"All our measurements are summarised in Table 2 and plotted in **Erreur ! Source du renvoi introuvable.**10 as a function of the median diameter of the distribution of the physical diameter of the particles.."

**Q28 :** p. 15. Line 18: "model of reference in the environment" – again a loose formulation.

This is rephrased (p16 l 12 to 14) :

"It should be remembered that the *in situ* scavenging measurements (Volken and Shumann, 1993; Laakso *et al.*, 2003; Chate, 2005) are only compared to the Slinn model."

p. 15. Line 24: I suggest here to consider the other physical processes involved in collision for different particle sizes, such as Brownian motion, thermophoresis, diffusio-phoresis, electroscavenging, etc. Good reference for that might be the paper of Ladino et al., ACP, pp. 9745 (2013), and the book of Pruppacher and Klett (2010).

Q29 :

This remark is aborted in different parts of the article :

First in section 1,we introduced all the mechanisms involved in the collection of aerosol particles by drop and droplets. We add in the text (p4 line 1 to p5 line 5):

[revised manuscript text omitted]

We calculated the contribution of diffusiophoresis (with the model of Davenport and Peters, 1978). We add this contribution on a new plot (figure 12):

[Figure]

Figure 3. Collection efficiencies measured in this study and by Lai *et al*. (1978). Both measurements are compared to Slinn (1977) and Beard (1974) models. The contribution of diffusiophoresis in both studies are computed following the description of Davenport and Peters (1978)

Thus we show that diffusiophoresis is a second order mechanism in our experiments.

Finally, we also suggest that Brownian diffusion (that is not considered in the Beard model) could explain why when the integration of the Beard model over the size distribution of the aerosol particle during our measurements for the smallest particles (0.22 µm) underestimates our measurement (figure 11 ). (p 20 line 9 to 14)

"These differences could be attributed to the fact that, for these points, the resolution of equation (12) requires an extrapolation of Beard (1974) calculations beyond the size range he investigated (continuous line on Figure 11).

Moreover, for the collection efficiency measured for the finest aerosol particles $(d_{ap} = 0.22 \, \mu m)$, the discrepancy observed with the Beard model could also be explained by the hypothesis of the simulations. Indeed, the Brownian motion was neglected. This can be justified in the particle size range investigated; however, it is much less justified when extrapolating the simulations to finer aerosol particles. "

Q30 :

p. 17. Line 10: To be honest, I do not see the significant improvement. The difference between measurements and theory is still large. I suggest you to indicate a range of possible collection efficiencies as a shadowed area, for instance, in the Figure by calculating collection efficiencies corresponding to the smallest and largest aerosol particle at a given d_ap.

The sentence and the caption of the figure have been modified to enhance the clarity of the statement.

Indeed, as the particles are not perfectly mono-dispersed in our experiments(this is also the case in other experiment see Lai et al for example), we cannot directly compare our measurements to the Beard model (red curve in figure 11). We first need to integrate this model over the size distribution of the particle we measured in the aerosol chamber for the considered measurement (in mass, because with fluorescein spectroscopy we measure a mass). And then compare this integration of the Beard model (red dots) can be compared to our measurement points (blue dots). We observe a nearly perfect superimposition except for the first and last measurement points and we added discussions to explain that.

[Figure]

Figure 4. Integration of the Beard (1974) model over the particle size distribution of each of our experiments, for a drop of 1.25 mm diameter.

We note a significant improvement of the agreement between our measurements and the Beard (1974) model since it is integrated over the entire particle size distribution measured during our experiments in BERGAME (red dots on Figure 11).

p. 17. Line 18: I do not really get the point. Have you taken into account the rear capture or not? If not, is it possible to do that and modify the collection efficiency curve?

Q31 :

This sentence was inappropriately formulated. I mean to say that the Slinn model that doesn't take into account rear capture leads to error on the collection efficiency of two orders of magnitudes in the submicronic range.

We reformulate to p17 line 27 to 28:

"Indeed, the Slinn (1977) model which neglects rear capture underestimates the collection efficiency by two orders of magnitude in the submicronic range compared to Beard's model (1974)."

As the Slinn model is easy to use and shows good agreements with Beard model for aerosol particles greater than one micron, we want to produce an additional correlation based on the simulations of Beard (1974) to model the elementary collection efficiency induced by Rear capture (equation 14 of the article).

$$E_{Rear-capture} = \frac{1}{3 \times 10^7} Re_{drop} \times St_{ap}^{-1.23} \quad (14)$$

This new elementary collection efficiency due to Rear capture would we added to the others presented in table 1 of the article. Thus Slinn model (with impaction, interception and Brownian motion) plus this new elementary mechanism for rear capture is in line with the Bear results, which we just validated experimentally. In his publication Bear noticed that rear capture is the main mechanism for particles with a Stokes number smaller than $5 \times 10^{-2}$. To produce an ease of use correlation we gather all his simulations points in that stokes number ranger and apply a power law fit.

To better explain we added the underlines text.

"Based on these comparisons, we can consider that the Beard (1974) model is validated for addressing the collection of the aerosol particles of the accumulation mode by raindrops. Finally, it seems necessary to provide, to facilitate its use, an analytical expression to assess the contribution of the rear capture ($E_{Rear-capture}$) to the raindrop collection efficiency. Indeed, the Slinn (1977) model which neglects rear capture underestimates the collection efficiency by two orders of magnitude in the submicronic range compared to Beard's model (1974). Furthermore, Beard (1974) noticed from his theoretical simulations that rear capture plays a main role in collection efficiency for aerosol particles with a Stokes number smaller than $5 \times 10^{-2}$. Thus, to derive an analytical expression for the elementary collection efficiency resulting from rear capture alone ($E_{rear\ capture}$), we gather in **Erreur ! Source du renvoi introuvable.**13 the collection efficiencies numerically simulated by Beard (1974) for a Stokes number smaller than $5 \times 10^{-2}$ (crosses in Figure 13). These collection efficiencies are plotted as a function of the Reynolds number of the drops and the Stokes number of the particles.

[Figure]

Figure 5. Semi-empirical parametrization of rear capture.

This figure suggests that the Reynolds number of the drop and Stokes number of the aerosol particles are the two parameters influencing rear capture. The dependency on these two dimensionless numbers is physical as the Reynolds number of the drop $\left(Re_{drop}\right)$ reflects the intensity and the size of the areas of recirculating flow in its wake and the particle Stokes number $\left(St_{ap}\right)$ reflects the susceptibility of the particle to pass through the recirculating flow in the wake of the drop without being trapped.

Applying a power law fit to the simulations of Beard (1974) yields equation 14.

$$E_{Rear-capture} = \frac{1}{3 \times 10^7} Re_{drop} \times St_{ap}^{-1.23} \quad (14)$$

This correlation is presented in solid lines in Figure 13 and shows a satisfactory agreement with K.V. Beard's simulations (crosses) in the corresponding range of drop Reynolds number and particle Stokes number. However, it should be kept in mind that this relationship is only valid for drop Reynolds numbers larger than 20 (a 280 µm drop at its terminal velocity), since below this critical value there is no recirculating flow behind the drop (Le Clair *et al.*, 1972). Finally, this new contribution should be added to those presented in Table 1 for raindrops. "

**Q32 :** p. 18. Line 4: I do not see the relevance of this figure, and its connection to the present experiments. Again, how do you account for the rear capture to calculate the collection efficency?

I think I answered this question in the answer of previous question. As the Beard model seems validated from our measurements, we gather on this figure the simulation points from Beard publication, for which he noticed they were dominated by rear capture (Stokes number smaller than $5\times10^{-2}$) and we uses these points to derive an elementary collection efficiency correlation for rear capture.

I think that without any ease of use correlation like the one produced in equation 14 rear capture would continue to be neglected.

**Q33 :** p. 20. Line 7: Please use dot instead of comma for the numbers.

Corrected

**Q34 :** p. 21. Line 3: The dimension of the volume (and, consequently, its error) is meter cube, nor meter.

This was a typo. It is corrected (p25 line 29-30):

"The uncertainty in the volume of dissolution is very low, we estimate its maximum error ($EMT_{V_{sol}}$) to be one millilitre."